# Mitochondrial Ribosomal Protein MRPS15 Is a Component of Cytosolic Ribosomes and Regulates Translation in Stressed Cardiomyocytes

**DOI:** 10.3390/ijms25063250

**Published:** 2024-03-13

**Authors:** Florian David, Emilie Roussel, Carine Froment, Tangra Draia-Nicolau, Françoise Pujol, Odile Burlet-Schiltz, Anthony K. Henras, Eric Lacazette, Florent Morfoisse, Florence Tatin, Jean-Jacques Diaz, Frédéric Catez, Barbara Garmy-Susini, Anne-Catherine Prats

**Affiliations:** 1Institut des Maladies Métaboliques et Cardiovasculaires (I2MC), Unité Mixte de Recherche (UMR) 1297, Institut National de la Santé et de la Recherche Médicale (Inserm), Université de Toulouse, 31432 Toulouse, France; davidflorian31@hotmail.fr (F.D.); emilieroussel55@gmail.com (E.R.); dn.tangra27@gmail.com (T.D.-N.); francoise.pujol@inserm.fr (F.P.); eric.lacazette@inserm.fr (E.L.); florent.morfoisse@inserm.fr (F.M.); florence.tatin@inserm.fr (F.T.); barbara.garmy-susini@inserm.fr (B.G.-S.); 2Institut de Pharmacologie et Biologie Structurale (IPBS), Centre National de la Recherche Scientifique (CNRS), Université de Toulouse, 31077 Toulouse, France; carine.froment@ipbs.fr (C.F.); odile.schiltz@ipbs.fr (O.B.-S.); 3Infrastructure Nationale de Protéomique, ProFI, FR 2048, 31077 Toulouse, France; 4UMR 5077 Molecular, Cellular and Developmental Biology (MCD), Centre de Biologie Intégrative (CBI), CNRS, Université de Toulouse, 31062 Toulouse, France; anthony.henras@univ-tlse3.fr; 5Centre de Recherche en Cancerologie de Lyon (CRCL), UMR 1052, Inserm, UMR 5286, CNRS, Centre Léon Bérard, Université de Lyon, 69008 Lyon, France; jeanjacques.diaz@lyon.unicancer.fr (J.-J.D.); frederic.catez@lyon.unicancer.fr (F.C.)

**Keywords:** ribosome heterogeneity, mitochondrial ribosomal protein, translational control, IRES, endoplasmic reticulum stress, cardiomyocyte, proteomics

## Abstract

Regulation of mRNA translation is a crucial step in controlling gene expression in stressed cells, impacting many pathologies, including heart ischemia. In recent years, ribosome heterogeneity has emerged as a key control mechanism driving the translation of subsets of mRNAs. In this study, we investigated variations in ribosome composition in human cardiomyocytes subjected to endoplasmic reticulum stress induced by tunicamycin treatment. Our findings demonstrate that this stress inhibits global translation in cardiomyocytes while activating internal ribosome entry site (IRES)-dependent translation. Analysis of translating ribosome composition in stressed and unstressed cardiomyocytes was conducted using mass spectrometry. We observed no significant changes in ribosomal protein composition, but several mitochondrial ribosomal proteins (MRPs) were identified in cytosolic polysomes, showing drastic variations between stressed and unstressed cells. The most notable increase in polysomes of stressed cells was observed in MRPS15. Its interaction with ribosomal proteins was confirmed by proximity ligation assay (PLA) and immunoprecipitation, suggesting its intrinsic role as a ribosomal component during stress. Knock-down or overexpression experiments of MRPS15 revealed its role as an activator of IRES-dependent translation. Furthermore, polysome profiling after immunoprecipitation with anti-MRPS15 antibody revealed that the “MRPS15 ribosome” is specialized in translating mRNAs involved in the unfolded protein response.

## 1. Introduction

The ribosome, has for a number of decades, been considered an apparatus able to translate genetic code without having an intrinsic regulatory capacity. Currently, several studies have modified this view of the translational machinery, and the concept of ribosome heterogeneity has appeared as a crucial mechanism in the process of translational control [1,2,3,4,5]. Ribosome heterogeneity includes variations of rRNA modifications [6,7]. Alterations in pseudo-uridylation and 2′O-methylation of rRNA have an impact on translation of mRNA coding transcription factors and growth factors and represent a relevant element in tumor biology [8,9,10,11,12]. Ribosome diversity is also conditioned by ribosomal protein (RP) stoichiometry and post-translational modifications [1,13]. Shi et al. have demonstrated the existence of ribosome heterogeneity in terms of ribosomal protein composition, which is responsible for a selective translation of subgroups of transcripts [1].

The first demonstration of ribosome specialization concerns RPS25/eS25: Landry et al. demonstrated that this ribosome component is not required for cap-dependent translation but is necessary for an alternative mechanism of translation initiation involving internal ribosome entry sites (IRESs) [14]. IRESs are mRNA structural elements that allow for internal ribosome recruitment instead of using the cap-dependent pathway that recruits the ribosome at the mRNA 5′ end [15]. This mechanism is crucial when cell global translation is blocked, which occurs in stress conditions [16]. The ribosomal protein RPS25/eS25 has been identified as an IRES-binding factor, suggesting the implication of specialized ribosomes for IRES-dependent translation in stress conditions [14,17]. This has been confirmed by Shi et al., who demonstrated that subpools of IRES-containing mRNA are translated by heterogeneous ribosomes [1]. This translational control mechanism is important in ischemic diseases where it increases translation of mRNA coding (lymph)angiogenic growth factors, which triggers tissue revascularization [18]. In an ischemic heart, this process also governs cardiomyocyte survival [19,20].

Mammalian cells contain a second class of ribosomes in mitochondria, called mitoribosomes [21,22]. These ribosomes are distinct from cytosolic ribosomes: their rRNA is encoded in the mitochondrial genome, while the 82 mitoribosomal proteins (MRPs) are encoded in the nucleus. Mitoribosomes specialize in synthesizing the 13 proteins encoded by the mitochondrial DNA.

Endoplasmic reticulum (ER) stress corresponds to the accumulation of misfolded or unfolded protein in the ER. Such accumulation induces the unfolded protein response (UPR), whose master regulator is the chaperone Glucose-regulated protein 78 (GRP78), a sensor of the 70 kDa heat shock protein (Hsp70) family. GRP78 is also known as immunoglobulin heavy chain binding protein (Bip) or heat shock protein family A member 5 (HSPA5). The GRP78/Bip mRNA was the first cellular mRNA in which an IRES was identified [23]. The UPR consists of a translational inhibition mechanism due to activation of PRK-like ER kinase (PERK), which phosphorylates the translation initiation factor eukaryotic translation initiation factor 2 (eIF2) subunit alpha. The second way is mediated by inositol-requiring-enzyme 1 (IRE-1), whose RNAse activity splices the X-box binding protein 1 (XBP1), resulting in expression of this transcription factor. The third way is mediated by the activation of transcription factor 6 (ATF6) [24]. GRP78 plays a key role: in normal conditions it is bound to PERK, eIF2 subunit alpha and ATF6 and inactivates these sensors. In stress conditions, it is overexpressed while being titrated by misfolded proteins, resulting in the release and activation of the three sensors [25]. ER stress occurs in several pathophysiological situations and diseases, including ischemic diseases [26,27]. In particular, myocardial infarction has been shown to activate ER stress [28]. This is revealed by a 2.5- to 10-fold increase in GRP78 and ATF6 expression and PERK activation. ER stress is accompanied by an arrest of global translation resulting from eIF2-α phosphorylation [16]. In contrast, ER stress has been shown to activate IRES-dependent translation [29,30]. In particular, the IRESs present in fibroblast growth factor 2 (FGF2) and vascular endothelial growth factor A (VEGFA) angiogenic factor mRNAs drive efficient translation when PERK is activated and eIF2-α phosphorylated.

In the present study, we aimed to identify variations of ribosome composition in cardiomyocytes in conditions of ER stress and to determine the impact of such variations on translation control. We investigated ribosome composition by a label-free quantitative shotgun proteomic approach in AC-16, a human cardiomyocyte cell line derived from ventricular heart tissue, subjected to tunicamycin treatment [31]. We studied the activity of IRESs present in mRNAs of FGF1, a major angiogenic growth factor, and insulin-like growth factor 1 receptor (IGF1R), a receptor involved in cardioprotection in ER stress conditions [32,33]. Mass spectrometry revealed few variations of RP expression by comparing polysomes of stressed and unstressed cells. In contrast, several MRPs showed drastic abundance variations and were found to be associated with cytosolic polysomes. By focusing on MRPS15, we showed that this protein is present in the cytosol of cardiomyocytes subjected to ER stress. MRPS15 interacts with the ribosome and is an activator of IRES-dependent translation, as well as of the translation of mRNAs involved in the unfolded protein response.

## 2. Results

### 2.1. Mitochondrial Ribosomal Proteins Are Differently Associated with Polysomes in Cardiomyocytes in Response to Stress

The first aim of our study was to analyze possible variations of ribosome composition in human cardiomyocytes in condition of ER stress. AC16 cells were thus treated with tunicamycin (tm), a classical inducer of ER stress, and a kinetic study was performed over 16 h (Figure 1a). We observed an increase of the ER stress marker GRP78 after 4 h of tm treatment with a significant upregulation after 8 h (Figure 1a, left and middle panels, Appendix A). The expected inhibition of global translation was analyzed using eIF2-α phosphorylation as a readout, showing an effect of tm on global translation efficiency as soon as after 2 h of treatment (Figure 1a, left and right panels, Appendix A). The condition of 4 h of treatment was selected to analyze ribosome composition. Polysomes and monosomes were separated on 10–50% sucrose gradients (Figure 1b, left panel). The polysome/monosome ratio decreased from 2.49 to 1.71 after tm treatment, confirming the impact of ER stress on the translational status of the cells (Figure 1b, right panel, Appendix A). Proteins from monosome or polysome fractions were then digested with trypsin and analyzed by nano-liquid chromatography–tandem mass spectrometry (nanoLC-MS/MS), leading to the identification and quantification of 4633 proteins. To evaluate protein changes, pairwise comparison based on MS intensity values were performed for each quantified protein between polysomes from tm-treated and untreated conditions. Variant proteins were selected based on their significant protein abundance variations between the two compared conditions (fold change (FC) >2 and <0.5, and Student’s *t* test *p* < 0.05) (see STAR method for details). Volcano plots, presented in Figure 1c, show variations of protein abundances in polysomes of tm-treated versus untreated cells (unt). Of these, 80 proteins were significantly more abundant and 115 less abundant in polysomes under stress, among them 27 proteins and 50 proteins which were found specifically in polysomes of stressed and unstressed cells, respectively (Figure 1c, Appendix A). Overall, no significant variation in RP level was observed between the two conditions. However, several RPs were not stoichiometric: RPS28/eS28 and RPLP2 were more abundant, while RPS27A/eS31, RPL36A/eL42 and RPL37/eL37 were less abundant in polysomes of both stressed and unstressed cells (Figure 2a,b, Appendix A). Unexpectedly, mitochondrial ribosomal proteins were detected in cytosolic polysomes and the stoichiometry of several ones showed significant variation with stress: MRPS15, MRPS33 and MRPL27 significantly increased, while MRPS35, MRPL33 and MRPL52 and MRPL54 decreased, in stressed cell polysomes (Figure 1c and Figure 2c).

These data suggest that mitochondrial ribosomal proteins are widely associated with cytosolic polysomes of ER-stressed as well as unstressed cardiomyocytes and that this association changes in stress conditions.

### 2.2. MRPS15 Is Partly Located in the Cytosol and Interacts with Ribosomal Proteins

To determine whether the MRPs identified above are a component of cytosolic ribosomes, we focused on MRPS15, one of the two MRPs that drastically increases in polysomes under stress (Figure 1c and Figure 2c). MRPS15 was first detected by capillary Simple Western (Figure 3a, Appendix A). Data showed a slight decrease of total MRPS15 (from 9.4 to 8.4 × 10^−4^ AU) in total cytoplasm after tm treatment, while in the cytosol only (after mitochondria depletion), a small increase of MRPS15 was detected (1.7 to 2.3 × 10^−5^ AU). MRPS15 intracellular localization was also analyzed by immunofluorescence with a confocal microscope. As expected, most MRPS15 co-localized with mitochondria (stained with Mitotracker), while a small portion of this protein was located in the cytosol (visible as a green signal in the merged panels) and significantly increased in tm-treated cells (Figure 3b, Appendix A).

The interaction of MRPS15 with ribosomes was analyzed by proximity ligation assay (PLA). RPS7/eS7 was chosen, as this RP is located on the surface of the ribosome [34]. As shown in Figure 3c, PLA using the antibodies anti-MRPS15 and anti-RPS7 revealed numerous fluorescent dots in the cytosol, whose number tended to increase in stressed cells (Figure 3c, Appendix A). These data showed a close interaction of MRPS15 with the ribosomal protein RPS7/eS7.

In addition, co-immunoprecipitation experiments were achieved with anti-MRPS15 antibody using cytosolic extracts and revealed by Simple Western with anti-RPS2 or anti-RPL10A antibodies. The results showed that MRPS15 is in the same complex as RPS2/uS5 and RPL10A/uL1 in the cytosol of stressed and unstressed cardiomyocytes (Figure 3d, Appendix A).

Altogether, these data demonstrated an interaction of MRPS15 with RPs of both the small and large ribosome subunits in AC16 cardiomyocytes, which was accentuated in polysomes in conditions of ER stress. This suggested that MRPS15 may be a component of cytosolic ribosomes in cardiomyocytes.

### 2.3. IRES-Dependent Translation Is Activated by ER Stress in Cardiomyocytes

In order to analyze whether ribosomes containing MRPS15 control translation of specific mRNAs, we focused our study on the IRES-dependent mechanism previously demonstrated to be activated during the early stress response while cap-dependent translation is blocked [15]. We analyzed the efficiency of IRES-dependent translation in AC16 cardiomyocytes subjected to different lengths of tm treatment, using the bicistronic vector assay validated previously [35]. The principle of this assay is that two cistrons, renilla luciferase (LucR) and firefly luciferase (LucF), are separated by an IRES. LucF reflects the IRES-dependent translation initiation, while LucR reflects the sum of the mRNA level and the cap-dependent initiation (Figure 4a). A lentivector containing the FGF1 IRES was used to transduce cardiomyocytes, and the LucF/LucR ratio reflecting the IRES activity was measured after tm treatment. Data showed a two-fold increase of IRES activity after 4 h of treatment (Figure 4b, Appendix A). The activity continued to increase up to 16 h. Consistent with the IRES activation, FGF1 protein expression measured by Simple Western increased after 4 h (Figure 4c). Similar data were obtained with the IGF1R IRES (Appendix A). These results show that IRES-dependent translation is triggered early upon ER stress in cardiomyocytes, as has been shown for hypoxia in a previous study [32,35].

### 2.4. MRPS15 Knock-Down Inhibits IRES Activation during ER Stress

The ability of MRPS15 to control IRES-dependent translation was investigated by RNA interference. MRPS15 knock-down was performed using a siRNA smartpool. AC16 cardiomyocytes were transduced with bicistronic lentivectors bearing either the FGF1 IRES or a hairpin control between the two luciferase cistrons. Cells were then transfected with siRNA against MRPS15. The maximal knock-down efficiency obtained after 48 h was only 20% (and not statistically significant), probably due to the requirement of MRPS15 in mitochondria (Figure 5a, Appendix A). The LucF/LucR ratio was significantly decreased in the same proportion in tm-treated AC16 cells, but not in untreated cells or cells transduced with the hairpin-containing bicistronic vector (Figure 5b, Appendix A). Furthermore, endogenous FGF1 expression also tended to decrease upon MRPS15 knock-down (Figure 5c, Appendix A). Despite the poor efficiency of the knock-down, these data suggest a role played by MRPS15 in the activation of IRES-dependent translation during ER stress.

### 2.5. Cytosolic MRPS15 Overexpression Promotes IRES Activation during ER Stress

To confirm the effect of MRPS15 on IRES-dependent translation and analyze its effect on different IRESs, we designed a lentivector producing a cytosolic form of MRPS15 devoid of the mitochondrial targeting sequence (Lenti-cMRPS15, Figure 6a). Transduction of AC16 cells with Lenti-cMRPS15 resulted in about 25% overexpression of MRPS15 on average compared to its endogenous level (Figure 6b, Appendix A). Cells transduced with a series of bicistronic lentivectors bearing the FGF1, IGF1R, VEGFD or EMCV IRES, or the hairpin control, respectively, were then transduced with Lenti-cMRPS15. IRES activities were measured, revealing that cMRPS15 overexpression is able to significantly promote FGF1 and IGF1R IRES activation but does not affect the VEGFD and EMCV IRESs (Figure 6c, Appendix A). This moderate but significant activation was observed only in tm-treated and not in untreated AC16 cells. These results confirm that MRPS15 has a positive effect on IRES activity, but suggest that this effect does not apply to all IRESs.

### 2.6. Ribosomes Containing MRPS15 Are More Associated with IRES-Containing mRNAs

In order to identify the mRNA families translated by ribosomes containing MRPS15, we performed a polysome-IP profiling experiment (Figure 7a). Polysomes of untreated or tm-treated AC16 cells were immunoprecipitated with anti-MRPS15 antibody (called MRPS15 polysomes below), followed by RNA sequencing. As a control, RNAseq was also performed on polysome-associated RNAs before immunoprecipitation, and the ratio elution/input was calculated to determine if specific mRNAs could be enriched after MRPS15 immunoprecipitation.

We first analyzed the variations in IRES-containing mRNAs in MRPS15-containing polysomes from untreated versus tm-treated AC16 cells (Figure 7b,c). Data show a variable fold change among most IRES-containing mRNAs detected in MRPS15 polysomes from untreated as well as tm-treated cells. In untreated cells, 15 mRNAs increased with a fold change factor ranging from 1.5 (LAMB1) to 6 (UTRN), while 9 mRNAs decreased between 1.6-fold (TP53) and 5.3-fold (BAX) (Figure 7b). In tm-treated cells, 16 IRES-containing mRNAs were more abundant in MRPS15 polysomes with fold change factors ranging from 1.4 (CDK1) to 13 (UTRN). 11 mRNAs decreased from 1.3 times (CDK2) to 6.2 times (BAX) (Figure 7c). The IGF1R mRNA increased 2 times in MRPS15 ribosomes from both untreated and tm-treated cells, in concordance with the regulation of its IRES activity by MRPS15 (Figure 6). In contrast, the FGF1 mRNA fold change decreased 2- and 2.7-fold in untreated and tm-treated cells, respectively. This discordance with the activation of its IRES activity by MRPS15 could result from the existence of several isoforms of endogenous FGF1 mRNA with different leader sequences [36].

Altogether, these data suggest that the MRPS15 ribosome regulates IRES-dependent translation. However, it is not a general activator of IRES-dependent translation during stress, since it shows specificity for a group of IRESs.

### 2.7. Ribosomes Containing MRPS15 Are Specialized in Translation of UPR mRNAs

Following the polysome-IP profiling performed above, we also analyzed mRNA families associated with MRPS15 polysomes in stressed versus unstressed cells. The resulting volcano plot shows more mRNAs whose association is increased by stress, up to 10-fold, than the opposite (Figure 8a and Appendix A). Among them, XBP1 mRNA, which is translated only under ER stress, is enriched 6-fold in the total polysomes, while it is enriched 9-fold in MRPS15 polysomes from tm-treated versus untreated cells (Appendix A). The HSPA5/GRP78/Bip mRNA, coding the key UPR sensor, is also enriched 6-fold in total polysomes and 11-fold in MRPS15 polysomes from tm-treated cells. The different mRNA families identified in MRPS15 polysomes are presented in a diagram (Figure 8b and Appendix A). Strikingly, all these families are linked to the unfolded protein response (UPR), the most enriched family being the ER stress response genes. The mRNA different families are represented in a schema summarizing the UPR (Figure 8c). These data clearly show that the MRPS15 ribosome specializes in the translation of the UPR mRNAs.

## 3. Discussion

The present study has led to a pivotal finding impacting translational response to ER stress, at least in human cardiomyocytes: an important number of mitochondrial ribosomal proteins are associated with translating ribosomes, and this association varies in stressed versus unstressed conditions. Such ribosome modifications govern translational reprogramming during ER stress: we show that MRPS15 is associated with cytosolic ribosomes and controls the translation of IRES-containing mRNAs and of mRNA families involved in the unfolded protein response.

One can ask whether these MRPs are intrinsic constituents of ribosomes incorporated during ribosome biogenesis or whether they bind to the cytosolic ribosomes. A previous publication has reported that MRPL18 is integrated into the 80S ribosome (co-immunoprecipitated with RPL4/uL4 and RPS6/eS6) [37]. This study by Zhang et al., performed in human HeLa cells and mouse embryonic fibroblasts, reports that a cytosolic form of MRPL18 is synthesized in heat shock conditions by a process of alternative initiation of translation at a CUG codon located downstream from the authentic AUG start codon. Such a mechanism would be possible in the case of MRPS15 mRNA, as a CUG codon is present at a position similar to that identified in the MRPL18 mRNA (Figure 6a). However, cytosolic MRPS15 could also originate from disaggregated mitochondria. The presence of MRPS15 in nuclei (but not specifically in nucleoli) is detected by immunofluorescence and rarely by PLA (Figure 3). Thus, we cannot exclude MRPS15 association with the ribosome prior to its export into the cytosol. An important novelty of our data compared to Zhang et al.’s study is that we detect an important number of MRPs bound to the polysomes and that a group of four MRPs are present in ribosomes of unstressed cells while nine other MRPs are associated with ribosomes of stressed cells [37]. This points out that the role of MRPs in cytosolic translation is broader than suggested by these authors.

The difference observed between the study by Zhang et al. and the present study raises the question of cell specificity of specialized ribosomes containing MRPs, as these authors worked with tumor and embryonic cells while we used cardiomyocytes. AC16 is an immortalized cell line derived from adult human ventricular cardiomyocytes, albeit keeping the cardiac myogenic cell markers [31]. Interestingly, a study performed in *Drosophila melanogaster* has reported that disruption of several RPs strongly alters heart function [38]. This study also shows that disruption of the mitochondrial MRPS33 induces severe cardiomyopathy. Such an observation suggests that MRPs may have a particular role in cardiomyocytes, in concordance with the wide incorporation into translating ribosomes revealed by the present data (Figure 1c).

MRPS15 have homologs not only in mammals but also in *Drosophila melanogaster*, *Caenorhabditis elegans* and *Saccharomyces cerevisiae* mitochondria [39]. This protein shares homology with the bacterial ribosomal protein S15, although it is considerably longer. The homologous part of MRPS15 shared with S15 is the central part of the protein. S15 interacts with the central domain of 16S rRNA and has a structural role in the ribosomal subunit [40]. The eukaryotic homolog of S15 is RPS13/uS15 [41,42]. Thus, we hypothesize that MRPS15 could be a paralog of RPS13/uS15, and we extrapolate that it might replace it in the MRPS15 ribosome, thus generating a change in ribosome structure that could affect it translational specificity.

We did not observe significant variation among RP stoichiometry between stressed and unstressed conditions from proteomic analysis (Figure 2a,b). However, several RPs diverged from stoichiometry, suggesting that they could be involved in specialized ribosomes. Previous reports support this hypothesis. RPS28/eS28 and RPLP2 are more abundant than other RPs in the present study. Interestingly, RPS28/eS28 age-downregulated variants have been described in Drosophila, generating a ribosomal heterogeneity that regulates the muscle proteome, while RPLP2 has been described as a translational activator for several viruses, including hepatitis B virus and coronavirus [43,44,45]. In contrast, RPS27A/eS31, RPL36A/eL42 and RPL37/eL37 are substoichiometric in our data. It has been reported that the amount of RPS27A/eS31 in ribosome, a protein involved in various pathologies including cancer and ribosomopathies, is regulated in response to DNA damage [46]. A study reported that the transcription of RPS27A/eS31 mRNA is increased in a p53-dependent manner, while a recent report showed that DNA double-strand breaks trigger the loss of this protein from ribosomes [47,48]. RPL36A/eL42 is upregulated in radioresistant tumors, while RPL37/eL37 expression increases in hypoxia and acidosis conditions [49]. All these observations suggest that the expression of RPs identified in our study is regulated and that they can contribute to ribosome heterogeneity, although we cannot rule out the possibility of an extraribosomal role.

In this study, ER stress was induced by tunicamycin, which generates protein misfolding by interfering with protein glycosylation [50]. It has been reported that tunicamycin-induced ER stress causes mitochondrial injury, potentially resulting in the release of MRPs into the cytosol, where these proteins, at least some of them, may play a role in translational reprogramming by binding to cytosolic ribosomes. We chose tunicamycin to induce ER stress in this study rather than dithiothreitol (DTT), a thiol antioxidant that activates the unfolded protein response by disrupting the oxidative protein folding environment [51]. DTT is known to have less specific effects than tunicamycin; in particular, it also activates the hypoxia response pathway and interferes with lipid metabolism [51,52]. The latter report recommends the use of tunicamycin rather than DTT to study the UPR [52]. In contrast to tunicamycin, DTT improves mitochondrial enzyme activity and attenuates mitochondrial dysfunction [53]. Kumar et al. even propose normalizing cardiac mitochondrial respiration in patients with end-stage heart failure by using DTT to reverse thiol oxidation [53]. Thus, studying the effect of several ER stress inducers having different impacts on mitochondria would be interesting for future investigations of MRP function in cardiomyocyte cytosol.

The effect of MRPS15 on IRES-dependent translation is significant only for FGF1 and IGF1R IRESs, but not for VEGFD and encephalomyocarditis virus (EMCV) IRESs. This suggests that the MRPS15 ribosome could be specific to a class of IRESs (see Figure 5 and Figure 6). However, the moderate effect observed on IRES activities is likely due to the inability to achieve efficient knock-down and the presence of large amounts of endogenous MRPS15 in the overexpression experiments. Although these data do not allow us to conclusively determine such specificity, they are consistent with our previous reports showing differential regulation of FGF, VEGF and EMCV IRESs in mouse HL-1 cardiomyocytes [32,35].

The data of polysome-IP profiling reveal the importance of MRPS15 on the translation of numerous endogenous mRNAs containing IRESs. It is notable that several mRNAs, such as utrophin (UTRN), X-linked inhibitor of apoptosis (XIAP), fragile X mental retardation protein (FMR1), apoptotic peptidase activating factor 1 (APAF1), hypoxia-inducible factor 1-alpha (HIF1A) and FGF2 mRNAs, are found to be enriched in MRP15 polysomes of both stressed and unstressed cells. This observation suggests that a fraction of MRPS15 ribosomes could be already efficient in translating these mRNAs in unstressed cells but would be more actively recruited in stress conditions. Identification of mRNA-coding proteins involved in apoptosis or hypoxic stress suggests that MRP15 impact in translational control might not be limited to ER stress but could be important for the translation of specific mRNAs during these two processes. Consistent with the effect of MRPS15 overexpression on IGF1R and VEGFD IRES activities (Figure 6), we observe that IGF1R mRNA is enriched in MRPS15 polysomes, while VEGFD mRNA is not detected among the enriched or depleted mRNAs. In contrast, the data are discordant for the FGF1 mRNA, which is depleted in MRPS15 polysomes, whereas its IRES activity is regulated by MRPS15 in the context of the dual luciferase bicistronic vector (Figure 6 and Figure 7). This inconsistency is explainable by the existence of several isoforms of endogenous FGF1 mRNA, resulting from the use of different promoters coupled with alternative splicing [36]. Only the mRNA isoform A contains the IRES tested in the present study, while the polysome profiling detects all isoforms.

A pivotal finding provided by the polysome-IP profiling of the present study is the broad role of the MRPS15 ribosome in the activation of translation of the mRNAs involved in the UPR pathways (Figure 8). For example, XBP1 and HSPA5/GRP78/Bip mRNAs are more enriched in the MRPS15 polysomes than in the total polysomes of stressed cells (Appendix A). Altogether, our data highlight a pivotal role of MRPS15 and presumably of other MRPs in translational control by specialized ribosomes during ER stress. This translational control is probably not limited to IRES-dependent translation: an IRES has been identified in the Bip mRNA [23], but this is not the case for all UPR mRNAs identified here. Translation regulation during ER stress is also mediated by upstream open reading frames (uORFs) [54,55]. Interestingly, Bip mRNA translation is also controlled by two uORF [56]. ATF4 mRNA, well known for its regulation mediated by uORFs but not by an IRES-dependent mechanism, is indeed enriched 3-fold in the polysomes of tm-treated cells, but this enrichment does not increase upon MRPS15 immunoprecipitation, suggesting that this mRNA is not translated by MRPS15 ribosomes. The same observation is made for CCAAT/enhancer-binding protein-homologous protein (CHOP)/ DNA damage-inducible transcript 3 (DDIT3) mRNA, containing a uORF, enriched almost 20-fold in the polysomes of tm-treated cells; however, there is no enrichment after MRPS15 immunoprecipitation [57]. These two cases suggest that MRPS15 ribosomes more readily control translation initiation mediated by IRESs than by uORFs. Further investigation is needed to fully understand the mechanism of action of MRPS15 in translation initiation.

## 4. Materials and Methods

All materials have been listed in a key resource table (Table 1).

### 4.1. Cell Lines

Female human embryonic HEK-293FT kidney cells (Invitrogen R700-07, Waltham, MA, USA), used to produce lentivectors, were cultivated in DMEM-GlutaMAX + Pyruvate (Life Technologies SAS, Saint-Aubin, France), supplemented with 10% fetal bovine serum (FBS) and MEM essential and non-essential amino acids (Sigma-Aldrich, St. Louis, MO, USA). Human ventricular AC16 cardiomyocytes (ATCC CRL-3568^TM^), used in all experiments, were cultivated in Dulbecco′s Modified Eagle′s Medium/Nutrient Mixture F-12 Ham containing 12% FBS, Penicillin/Streptomycin (100 U/mL–100 μg/mL) and 2 mM L-Glutamine. All cells were cultivated in a humidified chamber at 37 °C and 5% CO_2_. Endoplasmic reticulum stress induction medium was replaced by AC16 medium containing 2 µg/mL of tunicamycin. Dose and time of treatment were based on a previous study [29]. However, the appropriate conditions for treating AC16 cells were determined with a time course (Figure 1a and Figure 4c). IRES activation was the parameter that determined the time of stress. Increase of the ER stress marker GRP78 was the parameter used to choose the time of stress.

### 4.2. Bacterial Strains

Top 10 (InVitrogen, Thermofisher scientific C404003, Waltham, MA, USA) and Strataclone (Agilent Technologies, 200185, Santa Clara, CA, USA) *Escherichia coli* strains were used. Cells were stored at −80 °C and grown in LB medium at 37 °C. Top10 cells were used for plasmid amplification of pTRIP lentivector. Strataclone cells were used for recombination and amplification of PCR product into pSC-B-amp/kan plasmid.

### 4.3. Cell Transduction

For lentivector transduction, AC16 cardiomyocytes were plated into a T75 flask and transduced for 6 h in 7.5 mL of transduction medium (OptiMEM-GlutaMAX, Life Technologies SAS) containing 5 μg/mL protamine sulfate in the presence of lentivectors (MOI 5). AC16 cells were transduced with a 90% efficiency in mean.

### 4.4. Cell Transfection

siRNA treatment on transduced cells was performed after one cell passage in 24-well plates for reporter activity assay, 12-well plates for gene expression experiment, 6-well plates for protein expression experiments. AC16 were transfected by siRNAs as follows: one day after being plated, cells were transfected with 50 nM of small interference RNAs from Dharmacon ON-Target plus SMARTpool targeting MRPS15, or non-targeting siRNA control (siControl), using INTERFERin (Polyplus Transfection, Illkirch, France) according to the manufacturer’s recommendations, in Dulbecco’s Modified Eagle’s Medium/Nutrient Mixture F-12 Ham with FBS, L-Glutamine and penicillin-streptomycin. The media was changed 24 h after transfection, and the cells were incubated for 48 h at 37 °C before experiments.

### 4.5. Lentivector Construction

Bicistronic SIN lentivectors coding the renilla luciferase (LucR) and the stabilized firefly luciferase Luc+ (called LucF in the text) separated by an IRES or by the control hairpin have been described previously, except for the bicistronic construct with the IGF1R IRES [32,58]. The cDNA sequence of the human IGF1R IRES (1038 nt) was introduced between the two cistrons, in the pTRIP-DU3-CMV-MCS lentivector plasmid used above for the other IRESs [35,59]. The hairpin negative control contained a 63 nt-long palindromic sequence cloned between LucR and LucF genes. This control has been successfully validated in previous studies [35]. All cassettes were under the control of the cytomegalovirus (CMV) promoter. Plasmid construction and amplification was performed in the bacteria strain TOP10 (Thermofisher Scientific, Illkirch Graffenstaden, France).

### 4.6. Lentivector Production

Lentivector particles were produced using the CaCl_2_ method based on tri-transfection with the plasmids pTRIP (coding the genes of interest), pCMV-dR8.91 (coding lentiviral proteins) and pCMV-VSVG (coding envelope protein), CaCl_2_ and Hepes Buffered Saline (Sigma-Aldrich, Saint-Quentin-Fallavier, France) of HEK-293FT cells. The medium was replaced by OptiMem medium 6 h after tri-transfection. Viral supernatants were harvested 48 h after transfection, passed through 0.45 μm PVDF filters (Dominique Dutscher SAS, Brumath, France) and stored in aliquots at −80 °C until use. Viral production titers were assessed on HEK-293FT cells with serial dilutions of a lentivector expressing GFP and scored for green fluorescent protein (GFP) expression by flow cytometry analysis on a BD FACSVerse (BD Biosciences, Le Pont de Claix, France).

### 4.7. Reporter Activity Assay

For reporter lentivectors, luciferase activities in vitro and in vivo were performed using Dual-Luciferase Reporter Assay (Promega, Charbonnières-les-Bains, France). Briefly, proteins from AC16 cells were extracted with Passive Lysis Buffer (Promega France). Quantification of bioluminescence was performed with a luminometer (Centro LB960, Berthold, Thoiry, France) from biological replicates and with three technical replicates for each sample in the analysis plate.

### 4.8. Capillary Western

Cells were harvested on ice, washed with cold PBS and collected on RIPA buffer (Biobasic supplemented with protease inhibitor (Sigma, St. Louis, MO, USA), and after centrifugation at 13,000 rpm for 10 min at 4 °C, protein surpernatant was collected. Protein concentration was measured using a BCA Protein Assay Kit (Interchim, Montlucon, France). Diluted protein lysate was mixed with fluorescent master mix and heated at 95 °C for 2 min 30 s or 60 °C for 5 min, depending on the protein detected. A 3 μL quantity of protein mix (1 mg/mL maximal concentration) containing Protein Normalization Reagent, blocking reagent, wash buffer, target primary antibody (rabbit anti-MRPS15, diluted 1:50 (Abcam, Ab137070, Waltham, MA, USA), mouse anti-eIF2α diluted 1:50 (Cell Signaling, #2103, Danvers, MA, USA), Rabbit anti-Phospho-eIF2α (Ser51) diluted 1:50 (Cell Signaling, #9721); rabbit anti-FGF1 diluted 1:25 (Abcam Ab207321), rabbit anti-GRP78 diluted 1:250 (Novus biological, NB-56411, Centennial, CO, USA), Anti-IGF1R diluted 1:25 (Abcam, Ab182408), rabbit anti-RPS2 diluted 1:25 (Bethyl, A303-794A, Montgomery, TX, USA), mouse anti-RPS7 diluted 1:25 (Santa Cruz, Sc-377317, Santa Cruz, CA, USA), rabbit anti-RPS25 diluted 1:100 (Abcam, Ab254671), rabbit anti-RPL10A diluted 1:100 (Abcam, Ab240179), mouse anti-HSP60 diluted 1:25 (Santa Cruz, Sc-13115), secondary-HRP (ready to use rabbit of mouse “detection module”, DM-001 or DM-002), and chemiluminescent substrate were dispensed into designated wells in a manufacturer-provided microplate. The plate was loaded into the instrument (Jess, Protein Simple, San Jose, CA, USA), and proteins were drawn into individual capillaries on a 25-capillary cassette (12–230 kDa Separation Module SM-SW004, Protein Simple). Normalization reagent allows for the detection of total proteins in the capillary through the binding of amine group by a biomolecule and gets rid of a housekeeping protein that can harbor an inconsistent and unreliable expression. Graphs plotted in the figures below represent chemiluminescence values before normalization.

### 4.9. RNA Purification and cDNA Synthesis

Total RNA extraction from AC16 cells was performed using TRI Reagent according to the manufacturer’s instructions (Molecular Research Center Inc., Cincinnati, OH, USA). RNA quality and quantification were assessed by a Nanodrop spectrophotometer (Nanodrop 2000, Thermo Scientific, Waltham, MA, USA). A 750 ng quantity of RNA was used to synthesize cDNA from total RNA extract using a High-Capacity cDNA Reverse Transcription Kit (Applied Biosystems, Les Ulis, France), or 10 μL RNA was used with the immunoprecipitation extract (input and elution samples). Appropriate no-reverse transcription and no-template controls were included in the qPCR assay plate to monitor potential reagent or genomic DNA contaminations, respectively. The resulting cDNA was diluted 10 times in nuclease-free water for RNA from total RNA extract or 5 times for RNA from immunoprecipitation extract. All reactions for the PCR array were run in biological triplicates.

### 4.10. QPCR

A 2 uL quantity of cDNA was mixed with 2× ONEGreen FAST qPCR Premix (ozyme, OZYA008, Saint-Cyr-l’École, France) and 10 μM forward and reverse primers, according to manufacturer instruction. qPCR reaction were performed on Quantstudio 1 (Applied Biosystems). The oligonucleotide primers used are detailed above.

### 4.11. Polysomal RNA Preparation from Sucrose Gradients

AC16 cells were cultured in two 150 mm dishes. Ten minutes prior to harvesting, cells were treated with cycloheximide at 100 mg/mL. Cells were washed twice with PBS at room temperature containing 100 mg/mL cycloheximide and harvested with Trypsin. After centrifugation at 800 rpm for 5 min at 4 °C, cell pellets were washed in cold PBS containing 100 mg/mL cycloheximide; this step was performed twice. Cell pellets were then resuspended in hypotonic lysis buffer (10 mM HEPES-KOH pH 7.5; 10 mM KCl; 1.5 mM MgCl_2_) containing 100 mg/mL cycloheximide. Cells were centrifuged at 800 rpm for 5 min and resuspended in hypotonic lysis solution containing hypotonic buffer, 1 mM DTT, 0.5 U/mL RNasin and protease inhibitor 1×. After 20 min of incubation, cells were lysed using Dounce homogenizer (Thermo Fisher Scientific, Waltham, MA, USA). Cell lysates were centrifuged two times, the first at 1000× *g* for 10 min at 4 °C and the second at 10,000× *g* for 15 min; the supernatants (devoid of nuclei, organites and membranes) were collected and loaded onto a 10–50% sucrose gradient. The gradients were centrifuged in a Beckman SW41 Ti rotor at 39,000 rpm for 2.5 h at 4 °C with brake. Fractions were collected using a Foxy JR ISCO collector and UV optical unit type 11 (Teledyne ISCO, Lincoln, NE, USA). RNA was purified from pooled heavy fractions containing monosomes (fraction 7–11) or polysomes (fractions 12–19) as well as from cell lysate.

### 4.12. Protein Purification from Sucrose Gradient

Fractions corresponding to monosomes (fractions 7–11) or polysomes (fractions 12–19) were pooled, and 7.5 µL of glycogen (20 mg/mL) was added to fraction corresponding to monosomes and 12 µL to that corresponding to polysomes before TCA was added to form a 15% final concentration. After gentle homogenization, samples were incubated for 1:15 h on ice. Polysome and monosome fractions were centrifugated for 20 min at 13,000 rpm at 4 °C, the supernatant was removed, and 15 mL of acetone was added. Samples were centrifugated for 5 min at 13,000 rpm at 4 °C, the supernatant was removed, and the pellet was left to dry for 10 min at RT. The pellet was then resuspended in 1 mL of RIPA buffer.

### 4.13. Polysome Purification from Sucrose Cushions

AC16 cells were cultured in two 150 mm dishes. Ten minutes prior to harvesting, cells were treated with cycloheximide at 100 mg/mL. Cells were washed twice with cold PBS and scraped in 12 mL of cold PBS; 150 mm dishes were washed with 8 mL of cold PBS. Cells were centrifugated at 800 rpm for 5 min at 4 °C, and cell pellets were resuspended in cold PBS; this step was performed twice. Cell pellets were then resuspended in hypotonic buffer (10 mM Tris-HCl pH 7.4, 10 mM KCl, 1 mM MgCl_2_, 1× protease inhibitor, 2 mM vanadyl ribonucleoside complex) and centrifugated at 800 rpm for 5 min at 4 °C. Cell pellets were resuspended in hypotonic lysis solution containing hypotonic buffer and 1 mM DTT. After 20 min of incubation at 4 °C, cells were lysed using Dounce homogenizer, and cell lysates were centrifuged two times, the first at 1000× *g* for 10 min at 4 °C and the second at 10,000× *g* for 15 min. The supernatants were collected and loaded onto sucrose cushions (from bottom to top: 3 mL of sucrose 30%, 3 mL of sucrose 20%, 3 mL of sucrose 10%). Sucrose cushions were centrifuged in a Beckman SW41 Ti rotor at 39,000 rpm for 3 h at 4 °C with brake, and clear pellets were resuspended with hypotonic lysis solution.

### 4.14. Preparation of Cell Extracts with or without Mitochondria

AC16 cells were cultured and cell extracts prepared as for polysome purification (see above). Cell pellets were resuspended in hypotonic buffer (10 mM Tris-HCl pH 7.4, 10 mM KCl, 1 mM MgCl_2_, 1× protease inhibitor, 2 mM vanadyl ribonucleoside complex) and centrifugated at 800 rpm for 5 min at 4 °C. Cell pellets were resuspended in hypotonic lysis solution containing hypotonic buffer and 1 mM DTT. After 20 min of incubation at 4 °C, cells were lysed using Dounce homogenizer, and cell lysates were centrifuged two times, the first at 1000× *g* for 10 min at 4 °C (extracts with mitochondria) and the second at 10,000× *g* for 15 min (extracts without mitochondria).

### 4.15. In-Gel Trypsin Digestion and Mass Spectrometry Analysis

For mass spectrometry analysis, pooled fractions containing monosomes or polysomes (see above) prepared in quadruple biological replicates from AC16 cells treated or not with tunicamycin were subjected to TCA precipitation. Protein pellets were resuspended in 50 µL of RIPA buffer supplemented with 12.5 μL of Laemmli buffer 5× (1×: 40 mM Tris-HCL, 2% SDS, 24.6 mM DTT, 10% glycerol, 0.08% bromophenol blue) and were subjected to a disulfide bridge reduction for 10 min at 95 °C under agitation followed by an alkylation of cysteine residues in 60 mM iodoacetamide for 30 min in the dark at room temperature. Each reduced/alkylated protein sample was then digested using the S-Trap™ Mini spin column protocol [60]. Briefly, equivalent volumes of 10% SDS were added to each sample in order to reach a final SDS concentration of 5%. Undissolved matter was removed by centrifugation for 8 min at 13,000× *g*. Aqueous phosphoric acid (12%) was added at a ratio of 1:10 to the protein sample for a final concentration of ~1.2% phosphoric acid, followed by seven volumes of S-Trap binding buffer (90% methanol, 100 mM TEAB, pH 7.1). After gentle mixing, the protein solution was loaded into an S-Trap filter several times, each separated by a 4000× *g* centrifugation step, until all the SDS lysate/S-Trap buffer had passed through the column. Afterwards, the captured proteins were washed six times with 400 µL S-Trap binding buffer. Digestion was performed for 1 h at 47 °C and then overnight at 37 °C by the addition of 125 µL of trypsin (Sequencing Grade Modified Trypsin, Promega) at 8 ng/µL in 50 mM ammonium bicarbonate. The digested peptides were eluted by the addition of 80 µL of 50 mM ammonium bicarbonate and 1 min centrifugation at 4000× *g*, followed by 80 µL of 0.2% formic acid and 4000× *g* centrifugation (1 min) and finally 80 µL of 50% aqueous acetonitrile containing 0.2% formic acid and a last 1 min centrifugation step at 4000× *g*. The pooled eluates were dried, resuspended with 17 µL of 0.05% trifluoroacetic acid (TFA) in 2% acetonitrile (ACN) and sonicated for 10 min before analysis by online nanoLC using an UltiMate^®^ 3000 RSLCnano LC system (Thermo Scientific, Dionex) coupled to an Orbitrap Fusion™ Tribrid™ mass spectrometer (Thermo Scientific, Bremen, Germany) operating in positive mode. Five microliters of each sample was loaded in two injection replicates onto a 300 μm ID × 5mm PepMap C18 pre-column (Thermo Scientific, Dionex) at 20 µL/min in 2% ACN, 0.05% TFA. After 3 min of desalting, peptides were online separated on a 75 μm ID × 50 cm C18 column (packed in-house with Reprosil C18-AQ Pur 3 μm resin, Dr. Maisch; Proxeon Biosystems, Odense, Denmark) equilibrated in 90% of buffer A (0.2% formic acid (FA)), with a gradient of 10 to 30% of buffer B (80% ACN, 0.2% FA) for 100 min, then 30% to 45% for 20 min at a flow rate of 300 nL/min. The instrument was operated in data-dependent acquisition (DDA) mode using a top-speed approach (cycle time of 3 s). Survey scans of MS were acquired in the Orbitrap over 350–1400 *m*/*z* with a resolution of 120,000 (at 200 *m*/*z*), an automatic gain control (AGC) target value of be 4 × 10^5^ and a maximum injection time of 60 ms. The most intense multiply charged ions (2+ to 6+) per survey scan were selected at 1.7 *m*/*z* with quadrupole and fragmented by higher-energy collisional dissociation (HCD). The monoisotopic precursor selection was turned on, the intensity threshold for fragmentation was set to 25,000, and the normalized collision energy was set to 28%. The resulting fragments were analyzed in the Orbitrap with a resolution of 30,000 (at 200 *m*/*z*), an automatic gain control (AGC) target value of 5 × 10^4^ and a maximum injection time of 54 ms. Dynamic exclusion was used within 60 s with a 10 ppm tolerance, to prevent repetitive selection of the same peptide. For internal calibration, the 445.120025 ion was used as lock mass.

### 4.16. MS-Based Protein Identification and Label-Free Quantification

All raw MS files were processed with Proteome Discoverer software (version 2.3, Thermo Fisher Scientific) for database search with the Mascot search engine (version 2.6.2, Matrix Science, London, UK) combined with the Percolator algorithm (version 2.05) for PSM search optimization. Generated peak lists were searched against the SwissProt database with taxonomy Homo sapiens (20,241 sequences), supplemented with frequently observed contaminant sequences using a processing workflow consisting of the following parameters: mass tolerances in MS and MS/MS were set to 10 ppm and 20 mmu, respectively. Enzyme specificity was set to trypsin/P, and a maximum of three missed cleavages were allowed. Carbamidomethylation of cysteine was set as a fixed modification, whereas acetylation (N-terminal protein), oxidation (M, P, R), phosphorylation (S, T), methylation (K, R) and ubiquitination (K) were set as variable modifications. The Percolator algorithm was used to calculate a q-value for each peptide-spectrum match (PSM); peptides and PSM were validated based on Percolatorq-values at a false discovery rate (FDR) set to 1%. FDR was estimated using a target-decoy approach using the reversed database. The dataset was then filtered using a consensus workflow consisting of the following parameters: only PSMs with rank 1 and Mascot ion score ≥ 13 were considered. Peptide identifications were grouped into proteins according to the law of parsimony and filtered to 1% FDR.

For label-free relative quantification across samples, MS features detection and the cross-assignment of MS/MS information between runs were performed (it allows for the assignment of peptide sequences to detected but non-identified features) using the default parameter sets of the Minora Feature Detector and Feature Mapper nodes, respectively. Each protein intensity was based on the sum of unique peptide intensities and was normalized across all samples by the highest total abundance. Missing values were replaced for each run with random values sampled from distributions centered around medians of detected values of replicates (replicate-based resampling imputation mode), and protein ratios were calculated as the median of all possible pairwise peptide ratios calculated between replicates of all connected peptides (pairwise ratio-based). For each pairwise comparison, an unpaired two-tailed Student’s *t*-test was performed, and proteins were considered significantly enriched when their absolute log2-transformed fold change was higher than 1 and their *p*-value lower than 0.05. The error rate is managed by adjusting *p*-value using the Benjamini–Hochberg method. To eliminate false-positive hits from quantitation, two additional criteria were applied: only the proteins with an adjusted *p*-value lower than 0.05 and quantified in a minimum of three biological replicates before missing value replacement for at least one of the two compared conditions were selected. Volcano plots were drawn to visualize significant protein abundance variations between the two compared conditions. These represent −log10 (*p*-value) according to the log2 ratio. Protein abundances were summarized in iBAQ values by dividing the protein intensities by the number of observable peptides in order to determine the protein stoichiometry [61,62].

### 4.17. Immunoprecipitation

AC16 cells were cultured in 150 mm diameter dishes. Cells were washed twice with cold PBS (4 °C) and scraped in 12 mL of cold PBS; 150 mm dishes were washed with 8 mL of cold PBS. Cells were centrifugated at 800 rpm for 5 min at 4 °C, and cell pellets were resuspended in cold PBS; this step was performed twice. Cell pellets were resuspended in hypotonic buffer (10 mM Tris-HCl pH 7.4, 10 mM KCl, 1 mM MgCl_2_, 1× protease inhibitor, 2 mM vanadyl ribonucleoside complex) and centrifugated at 800 rpm for 5 min at 4 °C. Cell pellets were resuspended in 1 mL of hypotonic buffer and incubated for 15 min at 4 °C, then cell suspension was completed with 0.7% of Triton X-100 and incubated for 5 min at 4 °C. Cell lysates were centrifuged two times, the first at 1000× *g* for 10 min at 4 °C and the second at 10,000× *g* for 15 min; the supernatants were collected. Supernatants were split into two portions, and 0.5 mL of hypotonic buffer, 0.9 mg of magnetic Dynabeads protein A (ThermoFisher, 10001D) and 10 µg of antibody (Rabbit anti-MRPS15 (Abcam, Ab137070 or rabbit serum IgG) was added and incubated for 1 h at 4 °C with agitation. Microcentrifuge tubes were loaded on a magnetic rack, supernatants were eliminated, and beads were washed three times with hypotonic lysis solution (hypotonic lysis buffer and triton 0.7%). Proteins were eluted by resuspending beads in RIPA buffer and incubated for 5 min at 5 °C, then 2:30 min at 65 °C. Beads were eliminated using a magnetic rack, and elution was collected in new microcentrifuge tubes and conserved at −20 °C. For RNA purification, Trizol was added to the magnetic beads and incubated for 5 min at room temperature before magnetic bead elimination using a magnetic rack. Elution was also collected in new microcentrifuge tubes and conserved at −20 °C for short-term storage or −80 °C for long-term storage.

### 4.18. Immunofluorescence

Cells were cultivated on glass cover slips. For mitochondria staining, cells were incubated in AC16 medium containing 100 nM Mitotracker (Cell Signaling, #9082) for 30 min at 37 °C prior to fixation. Cells were fixed by incubated cells in ethanol/acetic acid solution (95% ethanol, 5% acetic acid) for 15 min at −20 °C and were then washed three times with cold PBS. Cells were permeabilized with an incubation of 1 min in PBS-Triton 0.1%. They were washed three times and incubated in blocking solution (FBS 1%, BSA 2%) for 1 h at room temperature. Antibodies were diluted in blocking solution: rabbit anti-MRPS15 diluted 1:500, (Abcam, Ab137070), mouse anti-RPS7 diluted 1:100 (Santa Cruz, Sc377317), mouse anti-RPS13 diluted 1:50 (Santa Cruz, Sc-398690), before the addition of the cover slip, for 1 h at room temperature. Cover slips were washed three times and incubated for 30 min at room temperature with secondary antibody: secondary donkey anti-rabbit Alexa 488 (Jackson, 711-545-152, Lansing, MI, USA), secondary donkey anti-mouse Alexa 594 (Jackson, 715-585-150), secondary donkey anti-mouse Alexa 647 (Interchim, FP-SC4110) diluted at 1:300 in blocking solution. Cover slips were washed three times before being mounted on a drop of VECTASHIELD Mounting Medium containing Dapi deposited on microscope slides.

### 4.19. Proximal Ligation Assay

Cells were cultured on glass cover slips and fixed in ethanol/acetic acid solution (95% ethanol, 5% acetic acid) for 15 min at −20 °C. They were permeabilized with an incubation of 1 min in PBS-Triton 0.1%, washed three times and incubated in blocking solution (FBS 1%, BSA 2%) for 1 h at room temperature. Antibodies were diluted in blocking solution: (rabbit anti-MRPS15 diluted 1:500 (Abcam, Ab137070), mouse anti-RPS7 diluted 1:100 (Santa Cruz, Sc377317), mouse anti-RPS13 diluted 1:50 (Santa Cruz, Sc-398690) before being added on cover slip for 1 h at room temperature. For next step reagents from Duolink In situ Red Starter kit Mouse/Rabbit (Sigma, DUO92101) was used. Cover slips were washed three times in 1× buffer A and incubated with the PLUS and MINUS PLA probes diluted 1:5 in Duolink antibody diluent for 1 h at 37 °C in humidity chamber. Cover slips were washed three times in wash buffer A 1× and incubated with 1× Duolink ligation buffer containing ligase diluted 1:40 for 30 min at 37 °C in a humidity chamber. Cover slips were washed three times in 1× wash buffer A and incubated in 1× amplification buffer containing polymerase diluted 1:80 for 100 min at 37 °C in a humidity chamber. Cover slips were washed three times in 0.01× wash buffer B before being mounted on Duolink PLA Mounting Medium with Dapi deposited on microscope slides.

### 4.20. Statistical Analysis

All statistical analyses were performed using 2-way ANOVA, Kruskall–Wallis, or Mann–Whitney rank comparisons test calculated on GraphPad Prism 10.1.2 software depending on the n number obtained and the experiment configuration. Results are expressed as mean ± standard error of the mean, * *p* < 0.05, ** *p* < 0.01, *** *p* < 0.001, **** *p* < 0.0001.

**Table 1 ijms-25-03250-t001:** Key resources list.

Reagent or Resource	Source	Identifier
**Antibodies**		
Mouse anti-eIF2α	Cell Signaling	#2103
Rabbit anti-Phospho-eIF2α (Ser51)	Cell Signaling	#9721
Rabbit anti-GRP78	Novus Biological	NB-56411
Rabbit anti-MRPS15	Abcam	Ab137070
Rabbit anti-FGF1	Abcam	Ab207321
Anti-IGF1R	Abcam	Ab182408
Rabbit anti-RPS2	Bethyl	A303-794A
Rabbit anti-RPS25	Abcam	Ab254671
Rabbit anti-RPL10A	Abcam	Ab240179
Mouse anti-RPS7	Santa Cruz	Sc-377317
Mouse anti-RPS13	Santa Cruz	Sc-398690
Mouse anti-HSP60	Santa Cruz	Sc-13115
Rabbit IgG serum	Sigma	I5006-1MG
Rabbit detection module	Protein Simple	DM-001
Mouse detection module	Protein Simple	DM-002
Secondary streptavidine-HRP	Protein Simple	043-459-2
Secondary donkey anti-Rabbit Alexa 488	Jackson	711-545-152
Secondary donkey anti-mouse Alexa 594	Jackson	715-585-150
Secondary donkey anti-mouse Alexa 647	Interchim	FP-SC4110
**Bacteria and Virus Strains**		
Escherichia Coli Top10	InVitrogen	C404003
Escherichia Coli Strataclone	Agilent technologies	200185
**Cell lines**		
AC-16 cardiomyocytes	ATCC (Manassas, VA, USA)	CRL-3568^TM^
HEK293 FT	Invitrogen	R700-07
**Chemicals, Peptides and Recombinant Proteins**		
TRI-Reagent	MRC Inc (Houston, TX, USA)	TR118
Isopropanol	Sigma-Aldrich	33539
Ethanol	Sigma-Aldrich	32221
Cycloheximide	Merck (Lowe, NJ, USA)	239764-100MG
NP40 (IGEPAL 630)	Sigma-Aldrich	I8896
Magnesium chloride	Serva (Heidelberg, Germany)	39772
Potassium chloride	Prolabo (Fontenay-sous-Bois, France)	26764.298
Sodium chloride	Honeywell (Charlott, NC, USA)	31434
Sucrose	Sigma	S7903
Proteinase inhibitor cocktail	Sigma-Aldrich	P2714
RNAse inhibitor	AppliedBiosystem	N8080119
Acetic acid 100%	Prolabo	20 104.298
Vanadyl ribonucleoside complexe	Sigma	R3380
PhoSTOP	Roche (Basel, Switzerland)	04 906 837 001
Bovine serum albumin standard	Euromedex (Souffelweyersheim, France)	04-100-812-E
Sodium dodecyl sulfate 20 %	Biosolve (Dieuze, France)	0019812323BS
RIPA	BioBasic (New York, NY, USA)	RB4476
MitoTracker^®^Red CMXRos	Cell Signaling	#9082
**Critical Commercial Assays**		
High capacity cDNA Reverse transcription kit	Thermofisher	4368814
ONEGreen FAST qPCR Premix	Ozyme	OZYA008
EZ-10 Spin Column Plasmid DNA Miniprep Kit	BioBasic	BS413
13.2 mL, Open-Top Thinwall Ultra-Clear Tube, 14 × 89 mm	Beckman	344059
StrataClone Blunt PCR Cloning Kit	Agilent	240207
Duolink In situ Red Starter kit Mouse/Rabbit	Sigma	DUO92101
Dual-Luciferase^®^ Reporter Assay system	Promega	E1980
Jess or Wes Separation Module	ProteinSimple	SM-SW004
Fluorescent 5x Master Mix 1	ProteinSimple	PS-FL01-8
**Experimental Models: Cell Lines and Medium**		
293FT	Invitrogen	R700-07
AC16 Human cardiomyocyte cell line	Sigma	SCC109
Dulbecco′s Modified Eagle′s Medium/Nutrient Mixture F-12 Ham	Sigma	D6434
Fetal bovine serum	Sigma	ES-009-B
L-Glutamine	Sigma	TMS-002-C
Penicillin-streptomycin	Sigma	TMS-AB2-C
Opti-MEM, reduced serum, no-phenol red	Gibco	11058021
**Oligonucleotides**		
HPRT	F: 5′-TGCTTTCCTTGGTCAGGCAGT-3′	R: 5′-CTTCGTGGGGTCCTTTTCACC-3′
FLuc	F: 5′-GTGTTGTTCCATTCCATCA-3′	R: 5′-TTGGCGAAGAAGGAGAATA-3′
Rluc	F: 5′-GCCTGATATTGAAGAAGATATTG-3′	R: 5′-CCTTTCTCTTTGAATGGTTC-3′
FGF1 CDS	F: 5′-GCT GAA GGG GAA ATC ACC AC-3′	R: 5′-CCC GTT GCT ACA GTA GAG GAG-3′
FGF1A	F: 5′-CCT CCT TTT CTG GGA GGA CA-3′	R: 5′-C AGC TTC TGC AAT GTC CAC-3′
FGF2	F: 5′-TGGTATGTGGCACTGAAACGA- 3′	R: 5′-GCCCAGGTCCTGTTTTGGAT-3′
VEGFA	F: 5′-TGCTGTCTTGGGTGCATTGGA-3′	R: 5′-CCACTTCGTGATGATTCTGCC-3′
VEGFC	F: 5′-AAAGAAGTTCCACCACCAAAC-3′	R: 5′-AGGGACACAACGACACACTTC-3′
18S	F: 5′-CAACTAAGAACGGCCATGCA-3′	R: 5′-AGCCTGCGGCTTAATTTGAC-3′
MRPS15 primer 1	F: 5′-CAAGATCCGCAGTTATGAAGAACAC-3′	R: 5′-TCCTCTGGTCAATGCTCATTAGC-3′
MRPS15 primer 2	F: 5′-CGTGACCAAGAAGGCTCTGTG-3′	R: 5′-GCTGCAGCCTTTAAGGCTCT-3′
**Recombinant DNA**		
pTRIP-CRHL+	Sequence available on Dryad, (2)	https://doi.org/10.5061/dryad.nvx0k6dq7, accessed on 20 February 2024
pTRIP-CRF1AL+	Sequence available on Dryad, (17; 26)	https://doi.org/10.5061/dryad.nvx0k6dq7, accessed on 20 February 2024
pTRIP-CRIGL+		
pTRIP-CRhVDL+		
pTRIP-CREL+	Sequence available on Dryad, (13)	https://doi.org/10.5061/dryad.nvx0k6dq7, accessed on 20 February 2024
pCMV-dR8.91	Addgene (Watertown, MA, USA)	
pCMV-VSV-G	Addgene	
pTRIP MRPS15		
ON-TARGETplus Human MRPS15 (64960) siRNA–SMARTpool	Dharmacon (Cambridge, UK)	L-013609-02-0020
ON-TARGETplus Non-targeting Pool	Dharmacon	D-001810-10-20
**Software and Algorithms**		
Prism 7	Graphpad (Boston, MA, USA)	https://www.graphpad.com/scientific-software/prism/, accessed on 20 February 2024
Microsoft 365 (excel, word, powerpoint) version 16.82	Microsoft office (Redmond, WA, USA)	
FIJI 1.530	FIJI (Madison, WI, USA)	https://fiji.sc/, accessed on 20 February 2024
ImageJ version 1.53	ImageJ (NIH, Stapleton, NY, USA)	https://imagej.nih.gov/ij/download.html, accessed on 20 February 2024
Zen black/Blue edition version 2.3 SP1 FP3	Zeiss (Rueil Malmaison, France)	https://www.zeiss.fr/microscopie/produits/microscope-software/zen-lite.html, accessed on 20 February 2024
QuantStudio	Applied Biosystems	https://www.thermofisher.com/fr/fr/home/global/forms/life-science/quantstudio-3-5-software.html, accessed on 20 February 2024
Microwin 2000	Berthold	https://fr.freedownloadmanager.org/Windows-PC/MikroWin-2000.html, accessed on 20 February 2024
**Other**		
LSM780 Zeiss confocal microscope	Zeiss	-
Jess capillary western	Protein Simple	-
SW 41 Ti Swinging-Bucket Rotor	Beckman	331362
Optima XL-100K Ultracentrifugeuse	Beckman	-

## Figures and Tables

**Figure 1 ijms-25-03250-f001:**
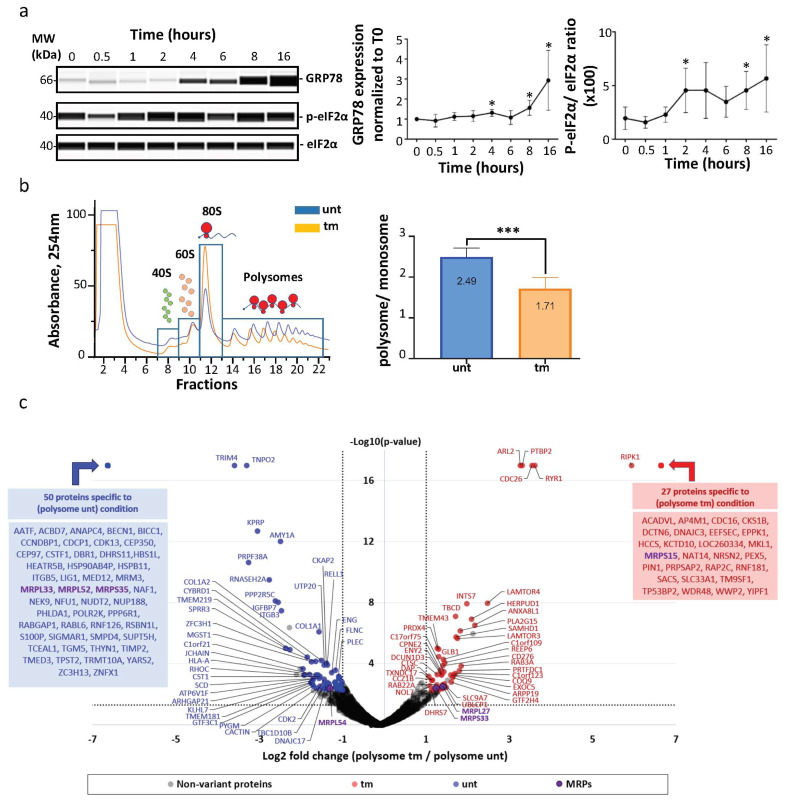
Polysome composition is modified under ER stress. (**a**) AC16 cardiomyocytes were subjected to tunicamycin treatment, and the ER stress time-course (0 h, 30 min, 1 h, 2 h, 4 h, 6 h, 8 h, 16 h) was carried out by analyzing eIF2α phosphorylation and GRP78 expression using capillary electrophoresis. GRP78 expression was quantified by normalization to 0 h time point (**middle panel**). EIF2α phosphorylation was quantified by measuring the ratio of P-eIF2α/total eIF2α for each time point (**right panel**). Experiments were reproduced 5 times and statistics performed with a non-parametric Mann–Whitney test, * *p* < 0.05 (Appendix A). (**b**) Proteins were purified from polysomal fractions of untreated or tunicamycin-treated (4 h) AC16 cardiomyocytes (Appendix A). Cytosolic lysate was purified on a sucrose gradient. P/M ratio (polysome/monosome) was determined by delimiting the 40S-60S-80S and polysome peaks and taking the lowest plateau values between each peak and by calculating the area under the curve (AUC). Then the sum of area values of polysome peaks was divided by the sum of the area of the 40S, 60S and 80S peaks. Experiments were reproduced 5 times, and the mean ± SEM is represented in the histogram. The P/M ratio of stressed cells condition was compared to unstressed condition using a non-parametric Mann–Whitney test, *** *p* < 0.001. (**c**) Proteins purified from polysomal fractions were identified and quantified using a label-free quantitative mass spectrometry approach (Appendix A). A volcano plot showing proteins significantly expressed in polysomal fractions of untreated cells (blue) and tunicamycin-treated cells (red) is presented. An unpaired bilateral Student’s *t*-test with equal variance was used. Variant significance thresholds are represented by an absolute log2-transformed fold change (FC) greater than 1 and a −log10-transformed (*p*-value) greater than 1.3 (dotted lines). The detailed list of proteins determined specifically detected in one of the conditions is given (blue and red boxes). Mitochondrial ribosomal proteins (MRPs) with significant variations are indicated in purple.

**Figure 2 ijms-25-03250-f002:**
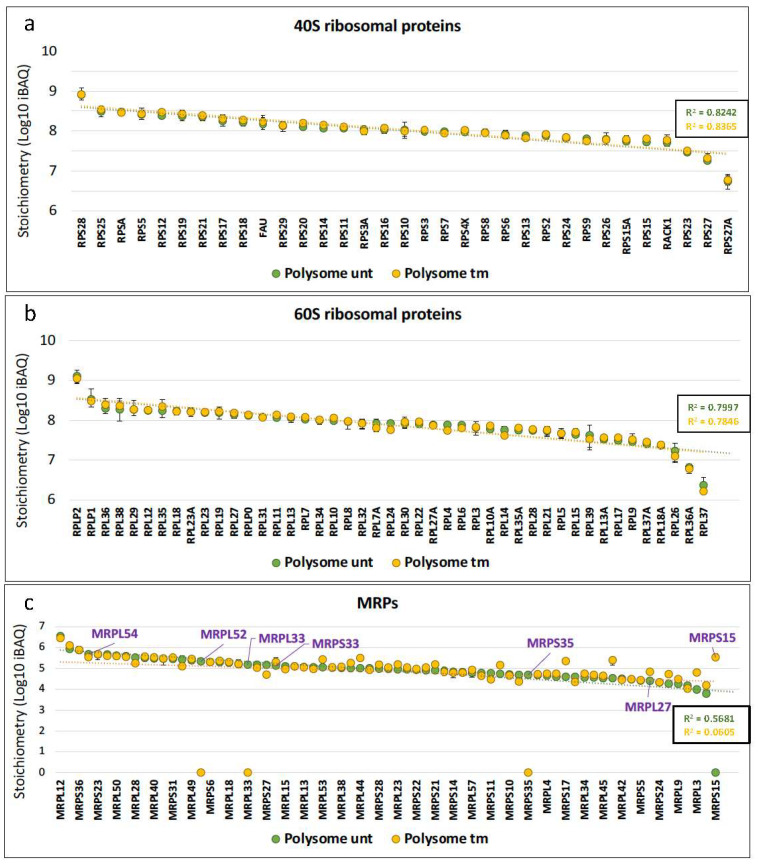
ER stress induces MRP stoichiometry variations in polysomal fractions. The relative stoichiometry of ribosomal proteins and mitochondrial ribosomal proteins in polysomes of AC16 cells was estimated from the data of Figure 1 by using the intensity-based absolute quantification (iBAQ) values in untreated (unt) versus tunicamycin-treated (tm) cell polysomes. The dotted line corresponds to the trend curve with the coefficient of determination (R^2^) in each condition. (**a**) 40S RPs, (**b**) 60S RPs, (**c**) mitochondrial ribosomal proteins. MRPs with statistically significant variation in the differential analysis are indicated in purple. On the abscissa axis MRPs are in the order provided in Appendix A, but only one out of two names are indicated for a better visibility (Appendix A).

**Figure 3 ijms-25-03250-f003:**
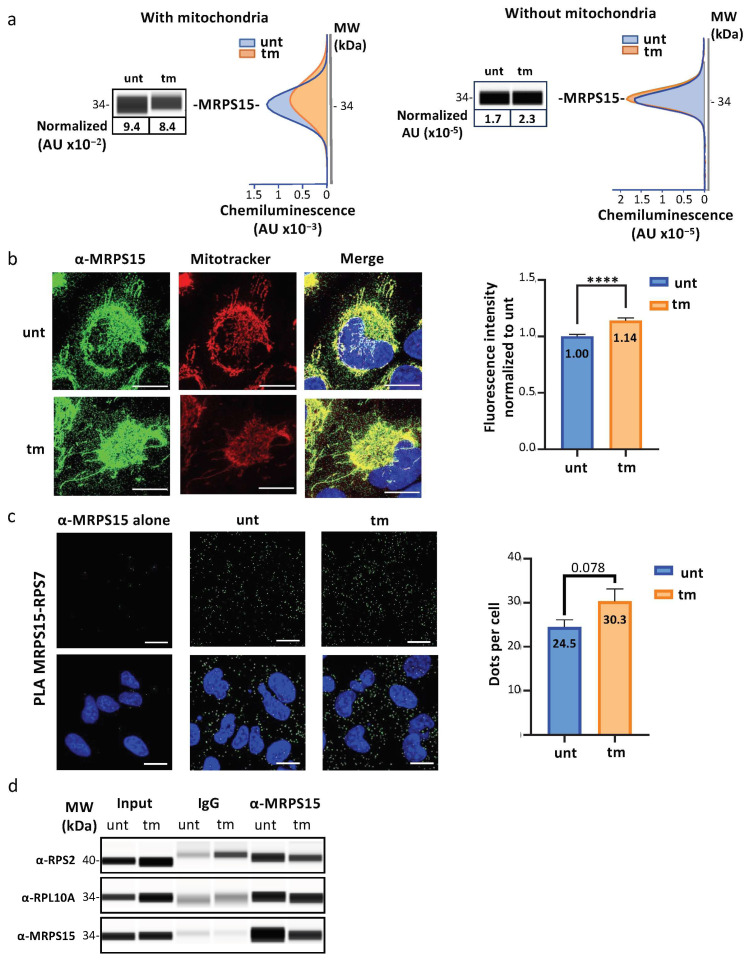
Identification of MRPS15 in the cytosol and interacting with ribosomal proteins. (**a**) MRPS15 quantification in AC16 total cytosolic extract (**left panel**) or cytosolic extract without mitochondria obtained by 10,000× *g* centrifugation for 15 min (see Section 4, **right panel**) in stressed and unstressed cells was performed using capillary electrophoresis Simple Western (Appendix A). (**b**) MRPS15 immunofluorescence staining in stressed and unstressed cell (green). Mitochondria were stained using Mitotracker (red), and a mask of mitochondria was carried out using threshold Otsu from ImageJ and was subtracted from the MRPS15 channel to quantify the MRPS15 staining. DAPI staining appears in blue. Quantification was performed on 3 independent experiments with at least 50 cells quantified per condition, and MRPS15 staining from the stressed cells was compared to unstressed cells with a non-parametric Mann–Whitney test (Appendix A). Bars correspond to 10 µM. **** *p* < 0.0001. (**c**) Proximal ligation assay using MRPS15 and RPS7 antibodies was performed on unstressed or stressed cells to visualize colocalization of MRPS15 with ribosomes (green). DAPI staining appears in blue. For unbiased analysis, dot quantification was performed after applying Otsu’s threshold method using ImageJ software version 1.53 and quantification carried out on 4 independent experiments with at least 12 images per condition and dots number in stressed cells was compared to unstressed condition using a non-parametric Mann–Whitney test (Appendix A). Bars correspond to 20 µM. (**d**) MRPS15 immunoprecipitation was performed using cytosolic extracts of unstressed or stressed cells without mitochondria. MRPS15 immunoprecipitation and RP co-immunoprecipitation were analyzed using capillary electrophoresis. Experiments were performed 3 times (Appendix A), and representative experiments are shown.

**Figure 4 ijms-25-03250-f004:**
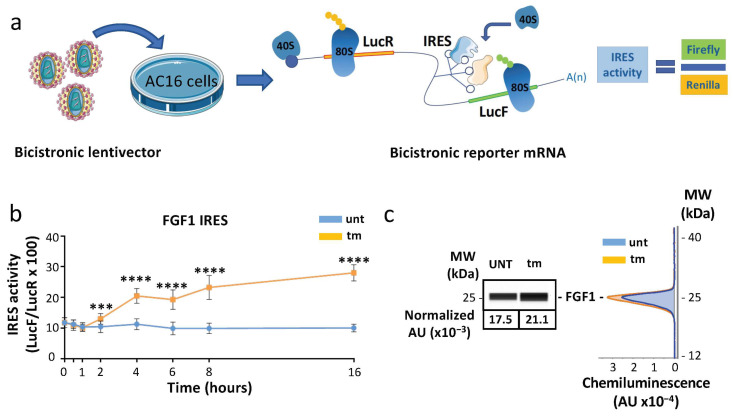
FGF1 IRES is activated after 4 h of ER stress in AC16 cells. (**a**) Schematic representation of the bicistronic reporter mRNA used for IRES activity experiments. Bicistronic lentivectors were used to transduce AC16 cardiomyocytes. The IRES is located between the two luciferase genes and drives the translation of the LucF cistron, while the LucR cistron is translated in a cap-dependent manner. IRES activity is expressed by the ratio LucF/LucR. (**b**) AC16 cardiomyocytes transduced with a bicistronic lentivector containing the FGF1 IRES were subjected to tunicamycin treatment. The FGF1 IRES activation time-course (0 h, 30 min, 1 h, 2 h, 4 h, 6 h, 8 h, 16 h) was carried out by measuring firefly/renilla ratio. Experiments were performed three times with 3 biological replicates (*n* = 9) and each time point was compared to 0 h time point with a non-parametric Mann–Whitney test (Appendix A); *** *p* < 0.001, **** *p* < 0.0001. (**c**) Endogenous FGF1 expression was measured by capillary electrophoresis, and a representative experiment from 9 independent experiments after 4 h of treatment is presented (Appendix A).

**Figure 5 ijms-25-03250-f005:**
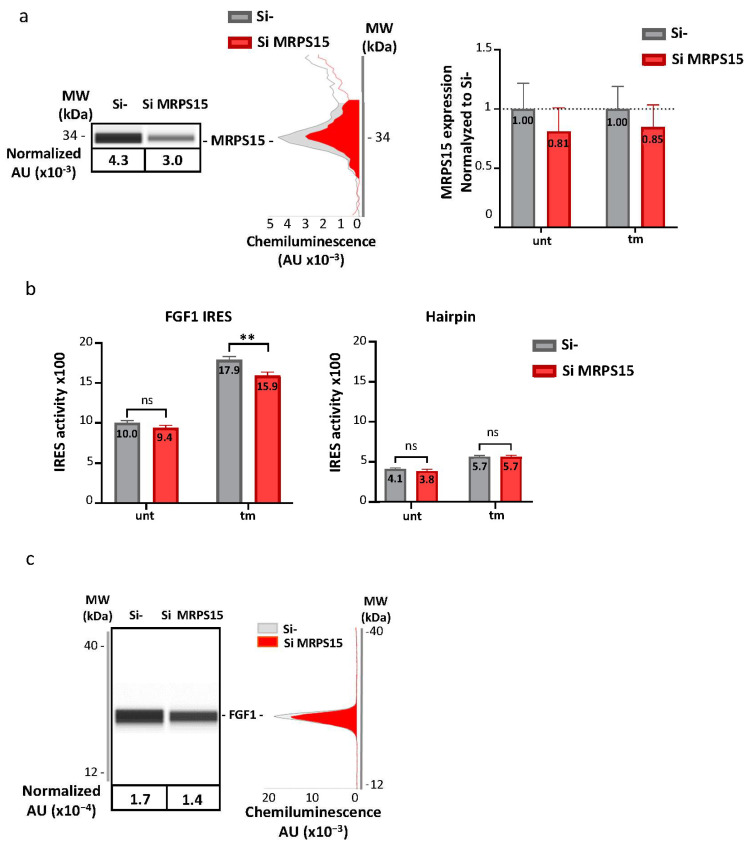
MRPS15 knock-down inhibits FGF1 IRES activity under ER stress. (**a**) AC16 cells were transfected with MRPS15 siRNA, and protein extinction was measured after 48 h by capillary electrophoresis Simple Western. A representative capillary electrophoresis and a histogram showing the mean ± SEM of 6 independent experiments are shown. (**b**) FGF1 IRES activity and hairpin control activity were obtained by measuring firefly/renilla ratio; the experiment was performed 4 times with 3 biological replicates (*n* = 12) for the FGF1 IRES and 3 times (*n* = 9) for the hairpin (Appendix A). Data were pooled and shown in a histogram corresponding to the mean ± SEM. Data of SiMRPS15 transfection were compared to corresponding Si control with a 2-way ANOVA test; ** *p* < 0.01, ns: non significant. (**c**) Endogeneous FGF1 expression was analyzed by capillary electrophoresis. A representative capillary electrophoresis and a histogram showing the mean ± SEM of 6 independent experiments are shown (Appendix A).

**Figure 6 ijms-25-03250-f006:**
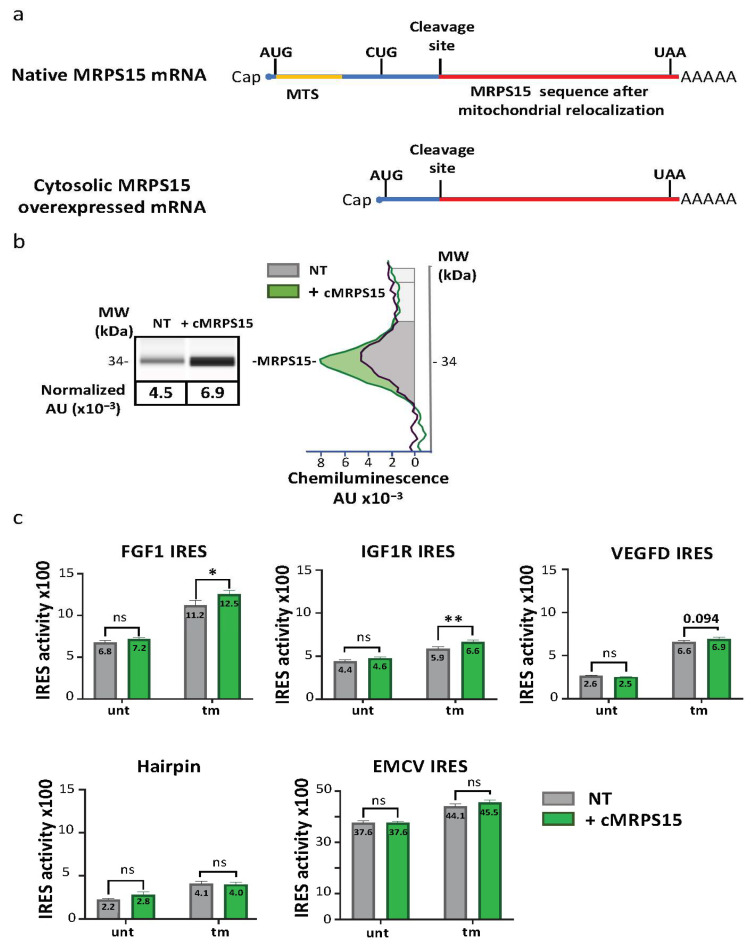
Cytosolic MRPS15 overexpression increases FGF1 and IGF1R IRES activities under ER stress. (**a**) Schematic representation of endogenous MRPS15 mRNA and of cytosolic MRPS15 mRNA used for cMRPS15 overexpression. A lentivector coding this cMRPS15 expression cassette was constructed and produced. (**b**) AC16 cells were transduced with the lentivector overexpressing cMRPS15 and with bicistronic IRES-containing lentivectors (FGF1, IGF1R, VEGFD, AMCV IRES of hairpin control). MRPS15 expression was measured by capillary electrophoresis; a representative experiment and a histogram showing the mean ± SEM of 3 independent experiments is shown. (**c**) IRES activity was obtained by measuring firefly/renilla ratio; each IRES experiment was performed at least 3 times with 3 biological replicates (*n* ≥ 9), and IRES activity of MRPS15 overexpression was compared to WT in corresponding conditions (untreated or treated tunicamycin) using a 2-way ANOVA test (Appendix A); * *p* < 0.05, ** *p* < 0.01, ns: non-significant.

**Figure 7 ijms-25-03250-f007:**
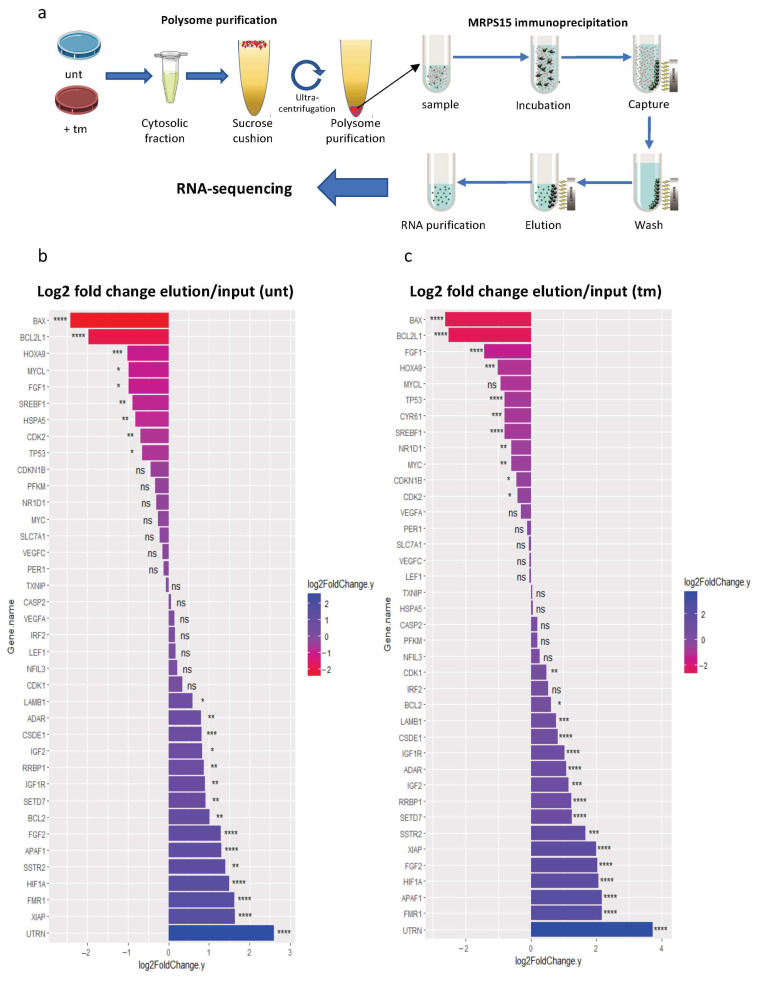
Ribosomes containing MRPS15 are more associated with IRES-containing mRNAs under ER stress. (**a**) Schematic representation of MRPS15 polysome immunoprecipitation profiling. (**b**,**c**) IRES-containing mRNA fold change (log2) of MRPS15 elution versus input samples from untreated (**b**) or tunicamycin-treated cells (**c**). Data represent 3 independent biological replicates for each condition (Appendix A); * *p* < 0.05, ** *p* < 0.01, *** *p* < 0.001, **** *p* < 0.0001, ns: non-significant.

**Figure 8 ijms-25-03250-f008:**
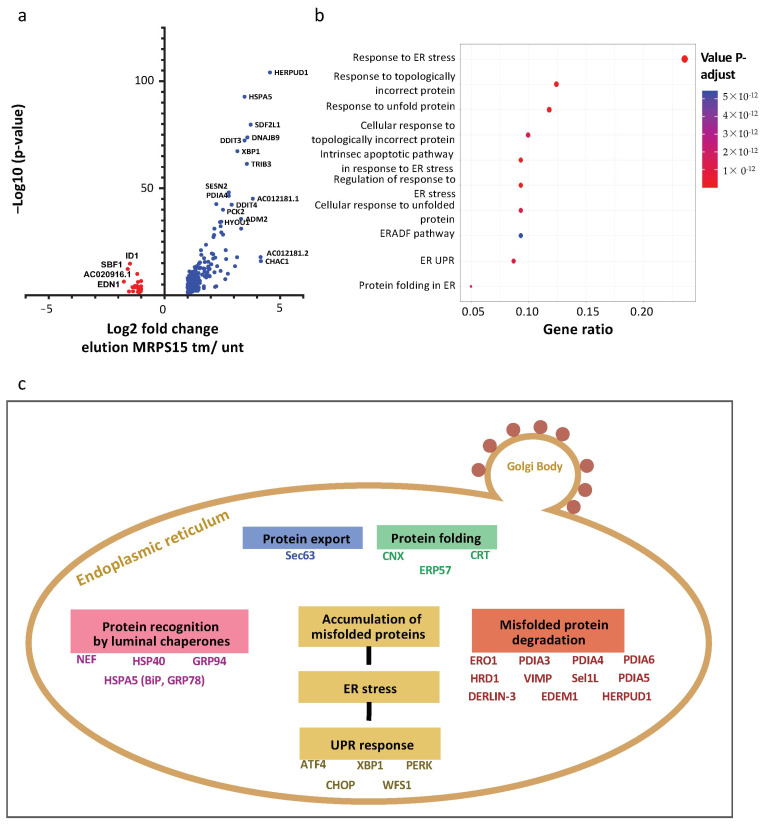
AC16 ribosomes containing MRPS15 are specialized in UPR mRNA translation. (**a**) Volcano plot from polysome-IP profiling showing RNAs significantly enriched in MRPS15 elution in tunicamycin-treated cells (blue) and MRPS15 elution in untreated cells (red). (**b**) The biological process of RNAs identified at a significant level in MRPS15 elution (tm-treated) was analyzed using enrichGO with R software (ClusterProfiler 4.10.0) and presented in a dot plot. (**c**) The most represented biological processes were analyzed using enrichKEGG with R software, and the complete pathway of proteins coded by mRNAs found significantly enriched in MRPS15 elution tm is shown.

## Data Availability

Mass spectrometry data are available via ProteomeXchange with identifier PXD050321. RNAseq IP data are available in ArrayExpress accession E-MTAB-13915.

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
