# Peer review of "Mitochondrial Ribosomal Protein MRPS15 Is a Component of Cytosolic Ribosomes and Regulates Translation in Stressed Cardiomyocytes"

_ijms, 2024, doi:10.3390/ijms25063250_

Round 1

Reviewer 1 Report

Comments and Suggestions for Authors

In the present study, the authors sought to investigate the variations of ribosome composition in human cardiomyocytes exposed to endoplasmic reticulum stress by using a proteomic approach, revealing the presence of several mitochondrial ribosomal proteins (MRPs) associated to cytosolic polysomes, with drastic variations in stressed condition compared to unstressed condition.

The topic is of physiopathological interest, the methodological approach appears appropriate; however, some important points should be improved before considering the paper suitable for publication.

Criticism and suggestions:

-        Abstract section should be improved. This section is missing of important information regarding methodological approach and the aim of the work appears confusing.

-        Introduction section appears confusing; the physiopathological role of reticulum stress and its role in I/R injury should be improved. The aims of the work should be reformulated.

-        Material and Methods section: dose and time of treatment of TM should be supported with specific references. Moreover, in this section the use of HEK-293FT cell line should be justified.

-        Results section is not presented clearly, please reformulate. Moreover, why the authors did not investigate others reticulum stress markers in WB section (such as ATF6, CHOP…)? Since TM is an important inducer of reticulum stress, would be interesting to include specific molecular markers to confirm the role of Mitochondrial ribosomal protein.

-        Discussion section should be improved, since suffer of shortcomings and the results should be discussed with more accuracy (for example the use of TM should be discussed etc…)

-        Other information should be included in figure legends, for example specifying cell lines used.

-        There are several typos in the text (for example the character style is not the same; see lines 37-39 page 1).

Author Response

Answer to reviewer 1:

We thank the reviewer for his/her constructive review that allowed us to improve the paper. Please find below the detailed answer to each comment. The modifications in the manuscript are indicated in red. Our answers to each comment are also indicated in red below.

In the present study, the authors sought to investigate the variations of ribosome composition in human cardiomyocytes exposed to endoplasmic reticulum stress by using a proteomic approach, revealing the presence of several mitochondrial ribosomal proteins (MRPs) associated to cytosolic polysomes, with drastic variations in stressed condition compared to unstressed condition.

The topic is of physiopathological interest, the methodological approach appears appropriate; however, some important points should be improved before considering the paper suitable for publication.

Criticism and suggestions:

-        Abstract section should be improved. This section is missing of important information regarding methodological approach and the aim of the work appears confusing.

The abstract has been modified and experimental details added lines 19-29 (we are limited to 200 words).

-        Introduction section appears confusing; the physiopathological role of reticulum stress and its role in I/R injury should be improved. The aims of the work should be reformulated.

A paragraph has been added lines 53 to 59, with three new references. The aims of the work have been reformulated lines 72-73.

-        Material and Methods section: dose and time of treatment of TM should be supported with specific references. Moreover, in this section the use of HEK-293FT cell line should be justified.

This has been indicated lines 439-443. Dose and time of treatment were based on a previous study (ref 20). However, the adequate condition for treating AC-16 cells was determined with a dose response (not shown) and a time course (Fig. 1a and 4c). Increase of the ER stress marker GRP78 was the parameter that determined the time of stress used in the study.

This has also be mentioned in the result section lines 92-93.

HEK-293FT is the cell line classically used to produce lentivectors. It has been mentioned lines 431-432.

-        Results section is not presented clearly, please reformulate. Moreover, why the authors did not investigate others reticulum stress markers in WB section (such as ATF6, CHOP…)? Since TM is an important inducer of reticulum stress, would be interesting to include specific molecular markers to confirm the role of Mitochondrial ribosomal protein.

Result section introduction has been rewritten lines 89 to 93. We agree that it would have been interesting to look at other markers of the unfolded protein response. However, the analysis was just aimed to check the efficiency of stress and we chose GRP78 as it has a central role in UPR by interacting with the three UPR proximal sensors IRE1, ATF6 and PERK. GRP78 is released in response to ER stress but its expression is also induced (and especially at the translational level by the IRES-dependent mechanism). It is classically used to detect ER stress. In addition, we were focusing on translational control thus we analyzed the target of PERK, eIF2alpha. Unfortunately, we do not have any data with the other markers for these experiments.

-        Discussion section should be improved, since suffer of shortcomings and the results should be discussed with more accuracy (for example the use of TM should be discussed etc…)

Three paragraphs have been added in the discussion (in red), in particular about the use of tunicamycin. Lines 401-416.

-        Other information should be included in figure legends, for example specifying cell lines used.

Information about the cell line has been added in the figure legends. Please note that AC16 cardiomyocytes have been used in all experiments. HEK293 have been used only to produce lentivectors, as indicated in Materials & Methods.

-        There are several typos in the text (for example the character style is not the same; see lines 37-39 page 1).

It was just a problem of interlining. We have corrected the typo errors.

Reviewer 2 Report

Comments and Suggestions for Authors

The article “Mitochondrial ribosomal protein MRPS15 is a component of cytosolic ribosomes and regulates translation in stressed cardiomyocytes” by David et al. analyzed changes in ribosome composition following tunicamycin-induced ER stress in cardiomyocytes using mass spectrometry. The authors identified certain ribosomal proteins that changed their abundance in response to stress. Further, many mitochondrial ribosomal proteins were also found enriched/decreased in the polysomes from stressed cells. The authors focus on MRPS15 as its association with the polysomes drastically increased. The authors next investigate the distribution and interaction of MRPS15 and further show through MRPS15 siRNA depletion or cytosolic MRPS15 overexpression that MRPS15 regulates translation of specific IRES-containing mRNAs. Lastly, through MRPS15 polysome-IP followed by RNA-seq the authors show that many MRPS15 associated mRNAs regulated the unfolded protein response in cells.

In general, the article is well written and of interest to the scientific community. However, prior to publication, the following aspects must be corrected.

 Major revisions:

·       Figure 1a: Instead of normalizing to T0, the authors should plot a control using untreated cells. As the cells metabolize nutrients and grow, changes in eIF2alpha and GRP78 are not accounted for when normalizing to T0.

·       Figure 2a/b, 4b: The chemiluminescence distribution graph is useless. It would be better to include total values with error bars from multiple experiments to be able to conclude significances of the changes reported.

·       Figure 3d/e: MRPS15 immunoprecipitation experiment was performed using cytosolic extracts of unstressed or stressed cells without mitochondria. Why does the MRPS15 IP bring down much more MRPS15 in untreated cells than in tunicamycin-treated cells? Isn’t that contradictory to previous statements? Why is there no signal for MRPS15 and RPL10A in the input? Based on Figure 2, a signal is expected in these fractions. For RPS7, it looks like general stickyness, as the signal is higher in the IgG controls than the IP, and hence an interaction cannot be concluded. Further, no strong enrichment in MRPS15 upon tunicamycin is observed, contradicting the previous mass spectrometry results, and questioning the validity of the manuscript.

Minor revisions:

·       Figure 1 c: Why aren’t all differentially expressed MRPs shown in volcano plot? It would further emphasize those, specifically MRPS15, which the authors focus on.

·       The article appears to contain two different fonts (lines 36-38 and 65-67).

·       Lines 126-129: The interaction of MRPS15 with ribosome was analyzed by proximity ligation assay (PLA). As shown in Fig. 3c, PLA using antibodies anti-MRPS15 and anti-RPS7 revealed numerous fluorescent dots in the cytosol, whose number tended to increase in stressed cells (Fig. 3c, Table S4). – What is the rationale for using RPS7? The dots are barely visible in the images included in the manuscript. Please increase the signal.

·       Please include sample preparation methods for figures 2a and 2b (with/without mitochondria).

·       Lines 133-135: The results showed that MRPS15 also interacts with RPS2 and RPL10A in the cytosol of stressed and unstressed cardiomyocytes (Fig. 3d, Table S5a-c). The same experiment performed with polysomal fractions confirmed the interaction of MRPS15 with RPS7 and RPL10A. These two statements are incorrect. The results show that MRPS15 is in the same complex as RPS2, RPL10A, RPS7 and RPL10A, it does NOT show that MRPs15 interacts with these proteins.

·       Lines 238-240: IRES activities were measured, revealing that cMRPS15 overexpression is able to significantly promote FGF1 and IGF1R IRES activation but does not affect the VEGFD and EMCV IRESs. Why/why not? Is it due to MRPS15 itself or a potential ITAF, which the authors could speculate on in their discussion.

·       Figure 5a (right panel): The error bars are overlapping, indicating an insignificant level of MRPS15 depletion. Conclusions from figure 5b must be taken with a grain of salt as the levels of MRPS15 depletion might be insufficient to cause effects on measurements in 5b.

Comments on the Quality of English Language

The manuscript is written well, only minor editing will be needed, e.g.

·       Lines 59-62: These ribosomes are different of cytosolic ribosomes: their rRNA is encoded in the mitochondrial genome, while the 82 mitoribosomal proteins (MRPs) are encoded in the nucleus, translated in the cytoplasm, and imported into the mitochondria. Mitoribosomes are dedicated to synthesizing the 13 proteins encoded in the mitochondrial DNA.

Author Response

Answer to reviewer 2:

We thank the reviewer for his/her constructive review that allowed us to improve the paper. Please find below the detailed answer to each comment. The modifications in the manuscript are indicated in red. Our answers to each comment are also indicated in red below.

Comments and Suggestions for Authors

The article “Mitochondrial ribosomal protein MRPS15 is a component of cytosolic ribosomes and regulates translation in stressed cardiomyocytes” by David et al. analyzed changes in ribosome composition following tunicamycin-induced ER stress in cardiomyocytes using mass spectrometry. The authors identified certain ribosomal proteins that changed their abundance in response to stress. Further, many mitochondrial ribosomal proteins were also found enriched/decreased in the polysomes from stressed cells. The authors focus on MRPS15 as its association with the polysomes drastically increased. The authors next investigate the distribution and interaction of MRPS15 and further show through MRPS15 siRNA depletion or cytosolic MRPS15 overexpression that MRPS15 regulates translation of specific IRES-containing mRNAs. Lastly, through MRPS15 polysome-IP followed by RNA-seq the authors show that many MRPS15 associated mRNAs regulated the unfolded protein response in cells.

In general, the article is well written and of interest to the scientific community. However, prior to publication, the following aspects must be corrected.

 Major revisions:

  • Figure 1a: Instead of normalizing to T0, the authors should plot a control using untreated cells. As the cells metabolize nutrients and grow, changes in eIF2alpha and GRP78 are not accounted for when normalizing to T0.

Thanks to the reviewer we have detected a mistake in the text. eIF2a phosphorylation was not quantified by normalizing to T0 but by measuring the ratio P-eIF2a/total eIF2a for each time point. The figure legend has been corrected lines 155-157. One can notice that eIF2a phosphorylation is confirmed by the translation inhibition observed in the polysome profile after 4 hours of stress (Fig. 1b).

  • Figure 2a/b, 4b: The chemiluminescence distribution graph is useless. It would be better to include total values with error bars from multiple experiments to be able to conclude significances of the changes reported.

We guess that the reviewer refers to Figure 3 rather than Figure 2. The total values with means and SD are provided in Table S3 (Fig. 3a/b) which is cited in the figure legend, lines 184 and 190. We wish to keep the chemiluminescence graph as it marks the difference between a capillary Western and a classical Western. For Fig. 4b, a table showing the data of 9 independent experiments has been added in Table S6g and the figure legend modified line 240.

  • Figure 3d/e: MRPS15 immunoprecipitation experiment was performed using cytosolic extracts of unstressed or stressed cells without mitochondria. Why does the MRPS15 IP bring down much more MRPS15 in untreated cells than in tunicamycin-treated cells? Isn’t that contradictory to previous statements? Why is there no signal for MRPS15 and RPL10A in the input? Based on Figure 2, a signal is expected in these fractions. For RPS7, it looks like general stickyness, as the signal is higher in the IgG controls than the IP, and hence an interaction cannot be concluded. Further, no strong enrichment in MRPS15 upon tunicamycin is observed, contradicting the previous mass spectrometry results, and questioning the validity of the manuscript.

Regarding Figure 3d, we agree that one should expect more MRPS15 in the treated versus untreated cell samples. This experiment is probably not really quantitative, however shows the interaction of MRPS15 with RPL10A and RPS2.

We agree that it is abnormal to have no signal in the input in Figure 3e. Consequently, we have removed these data as they do not provide clear information.

Minor revisions:

  • Figure 1 c: Why aren’t all differentially expressed MRPs shown in volcano plot? It would further emphasize those, specifically MRPS15, which the authors focus on.

All differentially expressed MRPs, i.e. with significant variations between untreated cells and tunicamycin-treated cells, are indicated in purple in volcano plot. Concerning MRPS15, given that it is specific to the tunicamycin-treated cells and therefore fold change and p-value cannot be calculated, a maximum arbitrary value of log2 Fold change and -log10(pvalue) assigned to it by the Proteome Discoverer software. Thus, all specific proteins below this arbitrary maximum point, including MRPS15, are shown in the top right red box for tunicamycin-treated cells specific proteins and top right blue box for untreated cells specific proteins (with the title “27 proteins specific to polysome tm condition”).

  • The article appears to contain two different fonts (lines 36-38 and 65-67).

Checked.

  • Lines 126-129: The interaction of MRPS15 with ribosome was analyzed by proximity ligation assay (PLA). As shown in Fig. 3c, PLA using antibodies anti-MRPS15 and anti-RPS7 revealed numerous fluorescent dots in the cytosol, whose number tended to increase in stressed cells (Fig. 3c, Table S4). – What is the rationale for using RPS7? The dots are barely visible in the images included in the manuscript. Please increase the signal.

The signal has been increased in Figure 3c.

The rationale of using RPS7 is that this RP is located on the surface of the ribosome. A sentence has been added line 137, with a reference (31).

  • Please include sample preparation methods for figures 2a and 2b (with/without mitochondria).

This preparation method has been added in the Mat & Meth, lines 576 to 584.

  • Lines 133-135: The results showed that MRPS15 also interacts with RPS2 and RPL10A in the cytosol of stressed and unstressed cardiomyocytes (Fig. 3d, Table S5a-c). The same experiment performed with polysomal fractions confirmed the interaction of MRPS15 with RPS7 and RPL10A. These two statements are incorrect. The results show that MRPS15 is in the same complex as RPS2, RPL10A, RPS7 and RPL10A, it does NOT show that MRPs15 interacts with these proteins.

The sentence has been changed as recommended by the reviewer (line 144). The text corresponding to polysomal fractions has been removed.

  • Lines 238-240: IRES activities were measured, revealing that cMRPS15 overexpression is able to significantly promote FGF1 and IGF1R IRES activation but does not affect the VEGFD and EMCV IRESs. Why/why not? Is it due to MRPS15 itself or a potential ITAF, which the authors could speculate on in their discussion.

This has been discussed lines 393-400.

  • Figure 5a (right panel): The error bars are overlapping, indicating an insignificant level of MRPS15 depletion. Conclusions from figure 5b must be taken with a grain of salt as the levels of MRPS15 depletion might be insufficient to cause effects on measurements in 5b.

The lack of statistical significance has been mentioned lines 222 and 227. However one must note that the weakness of the knock-down probably leads to underestimate the effect. This has also been discussed lines 393-400. The absence of effect on VEGFD and EMCV IRES may be due the poor knock-down efficiency.

Comments on the Quality of English Language

The manuscript is written well, only minor editing will be needed, e.g.

  • Lines 59-62: These ribosomes are different of cytosolic ribosomes: their rRNA is encoded in the mitochondrial genome, while the 82 mitoribosomal proteins (MRPs) are encoded in the nucleus, translated in the cytoplasm, and imported into the mitochondria. Mitoribosomes are dedicated to synthesizing the 13 proteins encoded in the mitochondrial DNA.

This has been corrected lines 67-71 of the revised text.

Round 2

Reviewer 1 Report

Comments and Suggestions for Authors

The authors have replied satisfactorily however some things still need to be improved.

lines 53 to 59 added by authors needs to be clarified and other information regarding ischemic heart and ER stress should be added (lines 60-66).

Authors should specify the adequate conditions for treating AC-16 cells and dose response curve should be added as main data.

 Discussion section should be improved, and the results should be discussed with more accuracy (the Authors only added information about the TM).

Comments on the Quality of English Language

The English language can be improved

Author Response

  • We thank the reviewer for his/her constructive comments which allowed us to improve our paper.

Comments and Suggestions for Authors

The authors have replied satisfactorily however some things still need to be improved.

lines 53 to 59 added by authors needs to be clarified and other information regarding ischemic heart and ER stress should be added (lines 60-66).

  • The introduction has been completed lines 67 to 85.

Authors should specify the adequate conditions for treating AC-16 cells and dose response curve should be added as main data.

  • The adequate condition for treating AC-16 cells has been tested by measuring the activation of FGF1 IRES with different doses. We recognize that we should have perfomed a dose response by measuring the ER stress markers. However we kept the dose of 8 mg/ml used in a previous study. We realize that the choice of the dose could have been chosen with more accuracy but we have no other explanation. We join the data of the IRES activity in this letter but in our view it is not publishable. However the dose used in this study worked to stress AC-16 cells and to activate the IRES thus does not question the data of the paper. 

Discussion section should be improved, and the results should be discussed with more accuracy (the Authors only added information about the TM).

  • Discussion section has been completed lines 431 to 450 and 460 to 472. Several references have been added.

Comments on the Quality of English Language

The English language can be improved

  • We have improved the English language

Reviewer 2 Report

Comments and Suggestions for Authors

The authors of the manuscript by David et al. revised their manuscript taking the suggestions made by the reviewers into account. The revised manuscript has improved and can be accepted in this form with one minor English revision.

Comments on the Quality of English Language

Line 76: submitted to tunicamycin treatment - the authors probably meant subjected to tunicamycin treatment.

Author Response

  • We thank the reviewer for his/her constructive comments which allowed us to improve our paper.

Comments and Suggestions for Authors

The authors of the manuscript by David et al. revised their manuscript taking the suggestions made by the reviewers into account. The revised manuscript has improved and can be accepted in this form with one minor English revision.

Comments on the Quality of English Language

Line 76: submitted to tunicamycin treatment - the authors probably meant subjected to tunicamycin treatment.

  • This has been corrected.
